# Structural basis for metal ion transport by the human SLC11 proteins DMT1 and NRAMP1

Márton Liziczai [ID], Ariane Fuchs [ID], Cristina Manatschal [ID] [✉] & Raimund Dutzler [ID] [✉]

Iron and manganese are essential nutrients whose transport across membranes is catalyzed by members of the SLC11 family. In humans, this protein family contains two paralogs, the ubiquitously expressed DMT1, which is involved in the uptake and distribution of $Fe^{2+}$ and $Mn^{2+}$, and NRAMP1, which participates in the resistance against infections and nutrient recycling. Despite previous studies contributing to our mechanistic understanding of the family, the structures of human SLC11 proteins and their relationship to functional properties have remained elusive. Here we describe the cryo-electron microscopy structures of DMT1 and NRAMP1 and relate them to their functional properties. We show that both proteins catalyze selective metal ion transport coupled to the symport of $H^+$, but additionally also mediate uncoupled $H^+$ flux. Their structures, while sharing general properties with known prokaryotic homologs, display distinct features that lead to stronger transition metal ion selectivity.

The transition metals iron and manganese are important nutrients that have to be absorbed into our diet. Their uptake and distribution in the body are tightly regulated and mediated by proteins of the SLC11 family[1]. This protein family is expressed in all kingdoms of life to catalyze the transport of $Fe^{2+}$ and $Mn^{2+}$ across cellular membranes (refs. 2,3.). Metal ion transport is coupled to the symport of $H^+$ serving as an energy source to ensure the efficient concentration of these rare elements inside cells[4]. In humans, the SLC11 family consists of two paralogs, the ubiquitously expressed SLC11A2 or DMT1, and SLC11A1 or NRAMP1, whose expression is confined to macrophages and neutrophils[1]. Among the human family members, our understanding of DMT1 function and physiology is currently the most advanced. The protein resides in the apical membrane of enterocytes where it catalyzes the uptake of free ferrous iron into intestinal epithelia[1]. This process requires the selection of the rare metal ion from a large background of $Ca^{2+}$ and $Mg^{2+}$, which are both several orders of magnitude more abundant. To prevent a toxic accumulation of $Fe^{2+}$, the expression of DMT1 in enterocytes is controlled at the translational level, with its pathological overexpression leading to iron overload[5]. Within the body, DMT1 localizes to endosomes to mediate the release

of endocytosed $Fe^{2+}$ into the cytoplasm powered by the proton gradient that is established by V-type ATPases[1]. A similar process underlies the recycling of iron from senescent erythrocytes after their engulfment by macrophages[1,6,7]. In macrophages and neutrophils, the second paralog NRAMP1 also plays a role in the defense against microbial infections[8]. Residing in late phagosomes of macrophages and tertiary granulocytes of neutrophils, it depletes endocytosed microbes from essential metal ions, thereby preventing their proliferation. In that way NRAMP1 contributes to the antimicrobial defense referred to as nutritional immunity[9], which is underlined by its malfunction leading to a compromised resistance against intracellular infections[10]. Over the years, the functional properties of DMT1 were characterized by electrophysiology and uptake studies of radioactive substrates in *X. laevis* oocytes[4] and in mammalian cells. These studies have defined the properties of a protein that transports diverse transition metal ions including $Fe^{2+}$, $Mn^{2+}$, $Co^{2+}$, and the toxic metal ion $Cd^{2+}$, while efficiently discriminating against the alkaline earth metal ions $Ca^{2+}$ and $Mg^{2+}$ (ref. 4.). Although not transported, $Ca^{2+}$ was described to act as a low-affinity inhibitor at millimolar concentration[11]. As a secondary active transporter, DMT1 utilizes a proton gradient to concentrate iron inside

Department of Biochemistry, University of Zurich, Zurich, Switzerland. [✉]e-mail: c.manatschal@bioc.uzh.ch; dutzler@bioc.uzh.ch

cells beyond its electrochemical equilibrium. However, the coupling is not strict, as protons were also described to pass the protein without being coupled to metal ion transport and vice versa uncoupled metal ion transport was observed at neutral pH[12].

In contrast to the well-characterized DMT1, little functional data is available for NRAMP1. The protein was described in different studies to either operate as an H[+]-coupled metal ion symporter[13,14] or antiporter[15,16] with a recent study proposing a role in the release of Mg[2+] from phagosomes[17].

Unlike in mammalian transporters, the primary substrate of prokaryotic family members is Mn[2+] as iron is usually taken up by prokaryotes as complex bound to siderophores[2]. A detailed comprehension of transport mechanisms in the SLC11 family was obtained from studies of different prokaryotic homologs, which were used as model systems for structural and functional characterization[18,19]. These studies have defined the architecture of SLC11 proteins in different conformations of the transport cycle and have provided the basis for their interaction with transition metal ions[20–23]. The ions bind to a conserved site located in the center of the transporter that is alternately accessible from respective sides of the membrane in inward- and outward-facing conformations. Both extreme states transition via intermediate conformations, where the access to the binding site is occluded. The ability to occupy the metal ion binding site regulates transport, which, similar to DMT1, extends to Fe[2+], Mn[2+], Co[2+], and Cd[2+] and is coupled to the symport of H[+] (refs. 20–22). However, despite the abundance of structural and mechanistic information on prokaryotic homologs, the structures of mammalian family members have remained elusive.

In this study, we were interested in the structural basis of metal ion transport by the human SLC11 paralogs DMT1 and NRAMP1. Particularly, we have addressed the hypothesis that both family members would share general mechanisms with their prokaryotic homologs while displaying distinct features, such as the presence of uncoupled H[+] leaks and the inhibition by Ca[2+]. We show that DMT1 and NRAMP1 catalyze the transport of Mn[2+] and Fe[2+] after their reconstitution into proteoliposomes and that the uptake of metal ions is coupled to the cotransport of H[+]. Structures of both transporters determined by cryo-EM show distinct conformations that define their interaction with transported substrates and they reveal elements that distinguishes them from prokaryotic transporters.

## Results
### Reconstitution and functional properties
To characterize the structural and functional properties of human members of the SLC11 family, we have expressed SLC11A1 (NRAMP1) and SLC11A2 (DMT1) by transient transfection of HEK293 cells and purified both paralogs in the detergent dodecyl-maltoside (DDM) supplemented with cholesterol-hemisuccinate (CHS). After their reconstitution into proteoliposomes, we have assayed the ability of either protein to mediate the transport of the transition metal ions Mn[2+] and Fe[2+], using the fluorophore calcein. A neutral pH was chosen to optimize calcein sensitivity. In all cases, we observe a metal ion concentration-dependent quenching of the fluorophore trapped inside the liposomes as a consequence of their uptake (Fig. 1a–d, Supplementary Fig. 1a–f). In both paralogs, Mn[2+] transport saturates at low micromolar concentrations with apparent $K_m$ values of 36 and 5 μM for DMT1 and NRAMP1, respectively (Fig. 1a, b, Supplementary Tables 1 and 2). In case of Fe[2+], we find similar transport properties with somewhat lower $K_m$ values of 2.5 μM for DMT1 and 1.4 μM for NRAMP1, indicating a higher affinity for this transition metal ion, which is the primary substrate of both transporters in a physiological setting, although in case of NRAMP1, the differences in the $K_m$ values between both ions measured with this in vitro assay are not statistically significant (Fig. 1c, d, Supplementary Tables 1 and 2). The transport properties for the two transition metal ions resemble those of the

prokaryotic homolog EcoDMT when assayed under equivalent conditions, underlining the strong conservation between pro- and eukaryotic family members (Supplementary Fig. 1g, h, Supplementary Tables 1 and 2).

We have also investigated whether the addition of Cd[2+], which does not quench calcein fluorescence, would interfere with Mn[2+] uptake and found a nearly complete suppression of transport at low μM concentrations in both human paralogs. Although only providing indirect evidence, this result suggests that the two metal ions could compete for the same binding site and consequently slow down Mn[2+] transport (Fig. 1e). Such competition is supported by structures of prokaryotic homologs of the SLC11 family, where Cd[2+] was shown to occupy the same metal ion binding site as Mn[2+] and Fe[2+] (refs. 20,24.). Experiments with proteoliposomes containing DMT1 and the Cd[2+]-sensitive fluorophore Fura-2 exhibit a Cd[2+] concentration dependent increase in fluorescence, which confirms transport of this toxic metal ion at low μM concentrations, consistent with previous experiments[4,20,25] (Supplementary Fig. 1i). Likewise, we tested whether we would find a similar interference with Mn[2+] and Fe[2+] uptake by the alkaline earth metal ions Ca[2+] and Mg[2+], both of which were excluded as substrates of DMT1 in earlier investigations[4,11], whereas Mg[2+] was proposed to be transported by NRAMP1 in a recent study[17]. For Ca[2+], no pronounced inhibition of Mn[2+] or Fe[2+] transport was observed in case of NRAMP1, whereas a reduction of Mn[2+] and Fe[2+] transport becomes apparent for DMT1 at a Ca[2+] concentration of 1 mM, which conforms with previous observations by electrophysiology and radiotracer uptake where Ca[2+] was proposed as non-competitive inhibitor with low mM $K_i$ (Fig. 1f, Supplementary Fig. 1j)[11,26]. In contrast to Cd[2+], we were unable to detect Ca[2+] transport by DMT1 using the Ca[2+]-sensitive metal ion indicator Fura-2 (Supplementary Fig. 1k) confirming earlier studies[4,11]. For Mg[2+], we did not find an indication of its interference with Mn[2+] or Fe[2+] transport in any of the two human paralogs up to a concentration of 1 mM (Fig. 1g, h). Together, our experiments refute the alkaline earth metal ions Mg[2+] and Ca[2+] as transported substrates of human SLC11 family members in agreement with previous studies on DMT1 and its prokaryotic homologs.

After characterizing their metal ion selectivity in a reconstituted system, we have investigated the function of both paralogs as H[+]-coupled symporters. To this end, we have assayed the Mn[2+] dependent acidification of vesicles as a consequence of coupled H[+] transport using the fluorophore ACMA. In these experiments, we find a robust metal ion concentration-dependent quenching of the fluorophore, emphasizing that Mn[2+] transport is coupled to H[+] even at neutral pH (Fig. 2a–c, Supplementary Fig. 2a, b). In case of Fe[2+], the concentration-dependent acidification is less obvious, which could either reflect the limited sensitivity of the assay or a previously reported uncoupled transport of this metal ion at elevated pH[12] (Supplementary Fig. 2c). For both transporters we also find a metal ion independent acidification originating from an uncoupled H[+] leak, which in NRAMP1-containing proteoliposomes is already pronounced at neutral pH (Fig. 2a, b). This effect is small at pH 7.4 and strongly increases upon a drop of the external pH to 6.5 (Supplementary Fig. 2d). This uncoupled H[+] transport is reminiscent of an equivalent process that was previously proposed for DMT1 based on two-electrode voltage clamp electrophysiology experiments[12]. To demonstrate that the observed H[+]-leak is mediated by the two proteins, we have used the DMT1-inhibitor TMBIT[27] to characterize its ability to interfere with acidification. We have previously shown that this molecule binds to DMT1 and its prokaryotic homolog EcoDMT in an outward-facing conformation, thereby inhibiting metal ion transport[28]. Similar inhibition is also found with reconstituted proteins in our study. Upon addition to the outside of proteoliposomes, we found a TMBIT concentration-dependent attenuation of metal ion transport in DMT1 (Supplementary Fig. 2e) and an equivalent decrease of acidification of either DMT1 or NRAMP1 in absence of metal ions (Fig. 2d, e, Supplementary Fig. 2f). Our results

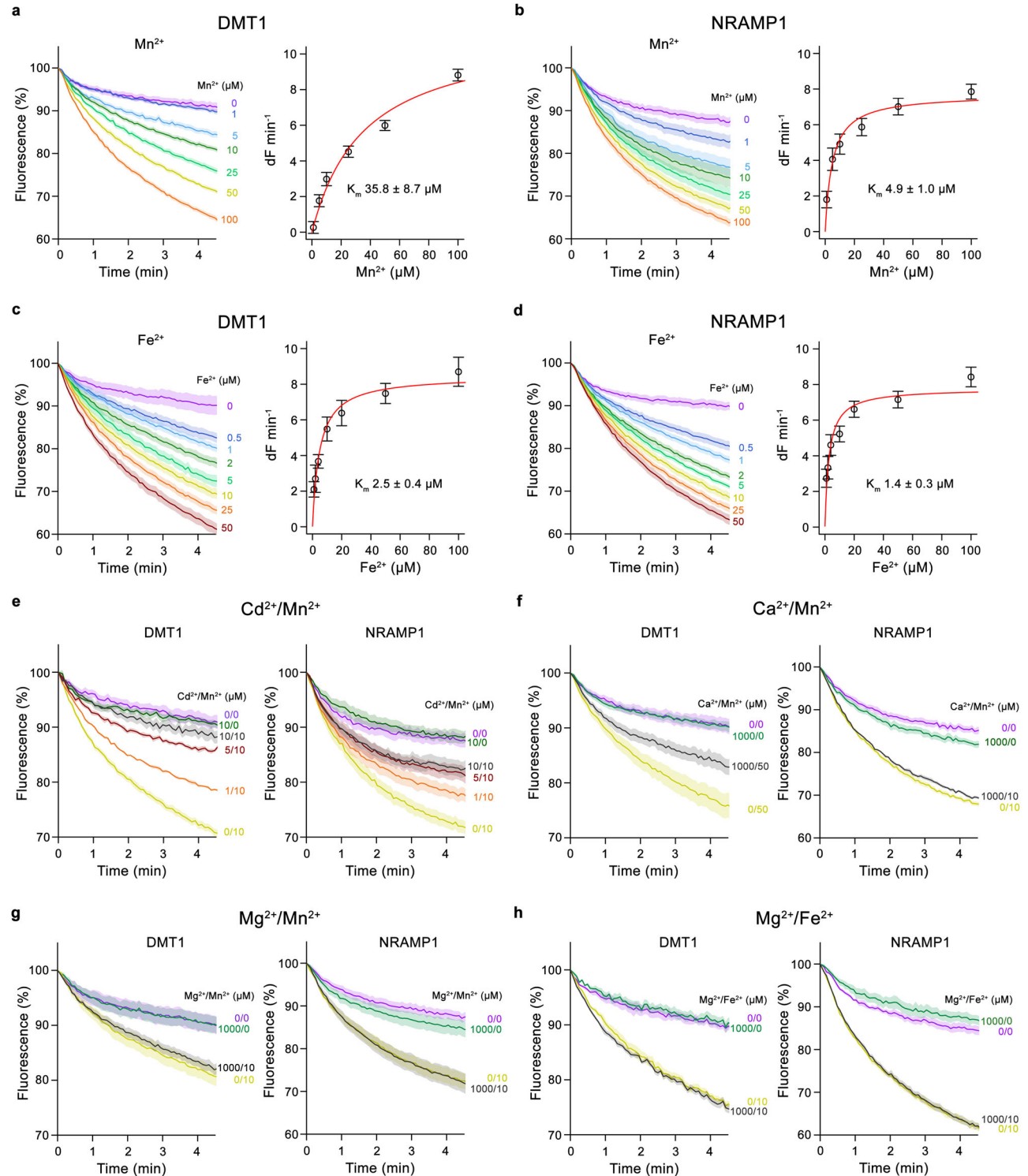

**Fig. 1 | Metal ion transport by human SLC11 paralogs. a–d** Transition metal ion transport into proteoliposomes by human SLC11 family members. Time and concentration dependence of metal ion uptake is shown on the left. Right panels show fits of the Michaelis-Menten equation to initial transport rates with respective $K_m$ values indicated (errors are s.d.) **a** DMT1 and **b** NRAMP1 mediated transport of $Mn^{2+}$; **c** DMT1 and, **d** NRAMP1 mediated transport of $Fe^{2+}$. **e** $Cd^{2+}$-concentration dependent inhibition of $Mn^{2+}$ transport into proteoliposomes containing DMT1 (left), or NRAMP1 (right). **f** Interference of $Mn^{2+}$ transport into proteoliposomes containing DMT1 (left), or NRAMP1 (right) by $Ca^{2+}$. **g** Interference of $Mn^{2+}$ transport into proteoliposomes containing DMT1 (left), or NRAMP1 (right) by $Mg^{2+}$. **h** Interference of $Fe^{2+}$ transport into proteoliposomes containing DMT1 (left), or NRAMP1 (right) by $Mg^{2+}$. **a–h** Uptake of $Mn^{2+}$ and $Fe^{2+}$ was assayed at pH 7.4 by the quenching of the fluorophore calcein trapped inside the vesicles. Data show the mean of four experiments from one reconstitution for $Mg^{2+}/Fe^{2+}$ competition assays, seven experiments from two independent reconstitutions for $Fe^{2+}$ transport by DMT1, eight experiments from two independent reconstitutions for all other experiments, errors are s.e.m. Fluorescence is normalized to the value after the addition of substrate (t = 0). Applied ion concentrations are indicated.

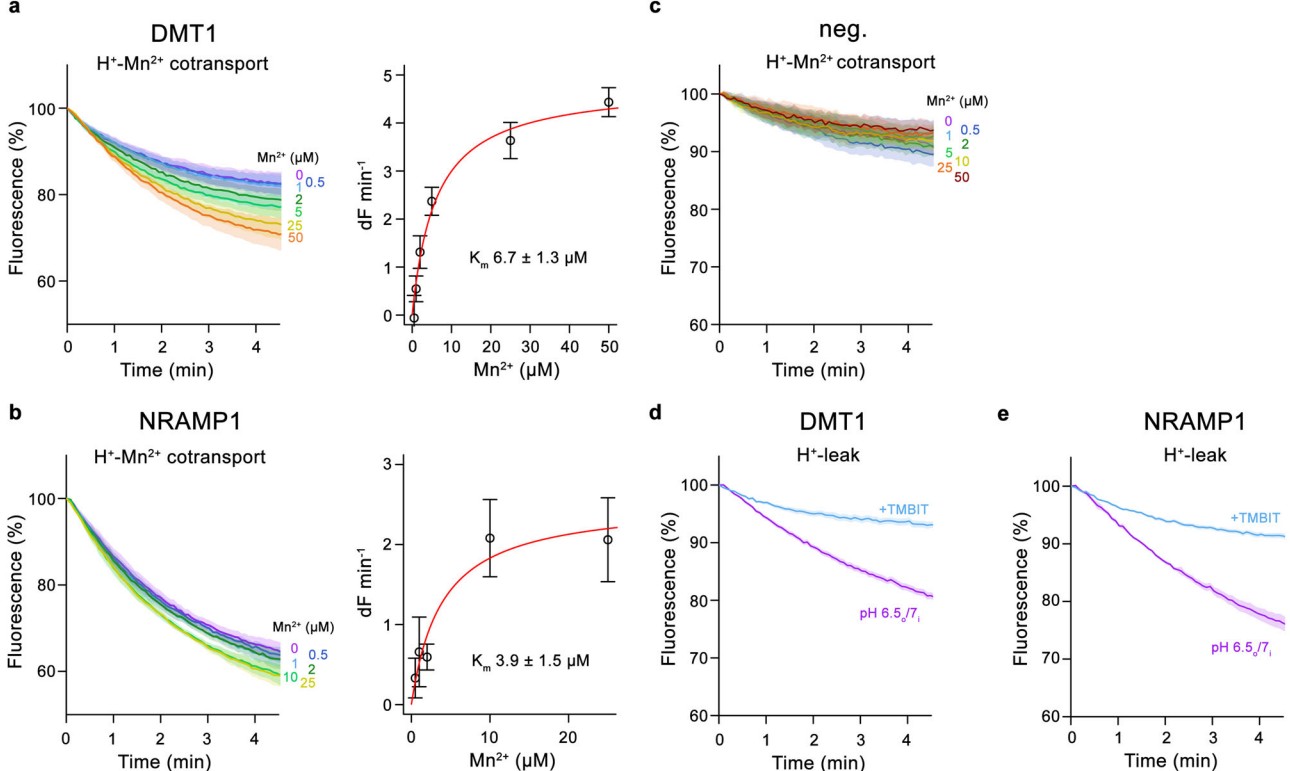

**Fig. 2 | H+ transport and its coupling to metal ions.** H+ transport mediated by human SLC11 paralogs assayed with the fluorophore ACMA. **a, b** Left, time and concentration dependent fluorescence decrease as a consequence of metal ion coupled H+ transport, assayed at pH 7.0 upon addition of $Mn^{2+}$ to proteoliposomes containing, **a** DMT1 or, **b** NRAMP1. Right, fits of a Michaelis-Menten equation to initial transport rates with respective $K_m$ values indicated. **c** Time and concentration dependent ACMA fluorescence decrease upon addition of $Mn^{2+}$ to proteoliposomes not containing a reconstituted transport protein is shown as control. **d, e** Metal ion independent H+ leak assayed at an asymmetric pH (outside pH 6.5, inside pH 7) and its inhibition upon addition of the DMT1 inhibitor TMBIT in, **d** DMT1 and **e**, NRAMP1. TMBIT was added at a concentration of 100 μM to the outside buffer. **a–e** Data show the mean of twelve experiments from two independent reconstitutions for the inhibition assay (**d, e**), the mean of seven and six experiments from two independent reconstitutions for the proton transport by DMT1 and NRAMP1, respectively (**a, b**), and the mean of four experiments from two independent reconstitutions for the negative control (**c**), errors are s.e.m. Fluorescence is normalized to the value after addition of substrate (t = 0). Applied ion concentrations are indicated.

thus demonstrate that the observed leak is mediated by either protein and that it can be suppressed by an inhibitor that locks the transporter in its outward-facing conformation.

## Structures of DMT1 and NRAMP1

As a next step, we explored the structural features underlying the described transport properties of both human paralogs by cryo-electron microscopy (cryo-EM). Due to the comparatively small size of these monomeric proteins (~52 kDa for their structured parts), which complicates particle alignment, we have initially engaged in the selection of nanobodies from synthetic libraries (termed sybodies)[29] against DMT1. These efforts have allowed us to identify two binders interacting with nanomolar affinities (of about 50 nM each) as determined by surface plasmon resonance spectroscopy (Supplementary Fig. 3a–e). The two proteins, Sb1DMT1 and Sb2DMT1 or short Sb1 and Sb2, bind to non-overlapping epitopes of DMT1 and do not interact strongly with its paralog NRAMP1 (Supplementary Fig. 3f-h). We then prepared samples of complexes of DMT1 containing either both binders (Sb1 and Sb2), or only Sb2 to record data in presence of $Mn^{2+}$ by cryo-EM. Datasets of the ternary DMT1/Sb1/Sb2 complex and the binary DMT1/Sb2 complex yielded reconstructions at resolutions of 3.9 and 3.6 Å respectively, which both permitted the assignment of structural features of the transporter (Fig. 3a, b, Supplementary Figs. 4 and 5, Table 1). Whereas both sybodies were visible in certain 2D classes, Sb2, which presumably binds to an intracellular epitope, was not defined

after 3D reconstruction due to the intrinsic heterogeneity of the interaction (Supplementary Figs. 4 and 5). The DMT1/Sb1/Sb2 complex thus displays the density of Sb1 bound to the extracellular side, bridging the loops connecting α1 and α2, α5, and α6, and the extended region between α7 and α8, whereas no sybody density was discernible in the DMT1/Sb2 complex (Fig. 3a, b). In both datasets, the transporter resides in a single conformation that carries features of an occluded state observed in homologs from prokaryotes and plants[22,23,30]. The only difference between the two DMT1 structures concerns a small conformational change in the extracellular part of α6a, whose full engagement with α1b was hampered by the bound Sb1 (Fig. 3c).

As in case of NRAMP1 the low yield of the protein in detergent solution has complicated sybody selection, we have attempted structure determination of this paralog without fiducial markers by combining 57'783 micrographs from five datasets. These data have allowed us to identify two distinct conformations of the transporter in the same sample (Fig. 3d–g, Supplementary Fig. 6, Table 1). One structure reconstructed at a nominal resolution of 3.9 Å is equivalent to the conformation observed for DMT1, as illustrated in the RMSD of 0.5 Å obtained from a superposition of Cα-atoms (Fig. 3d, e). The second structure at a nominal resolution of 3.7 Å (RMSD to DMT1 1.5 Å) is distinguished by pronounced differences in α1a and smaller differences in α4 and 5 (Fig. 3f, g). Together our structures of DMT1 and NRAMP1 display two states on the transport cycle of SLC11 proteins (Supplementary Fig. 7).

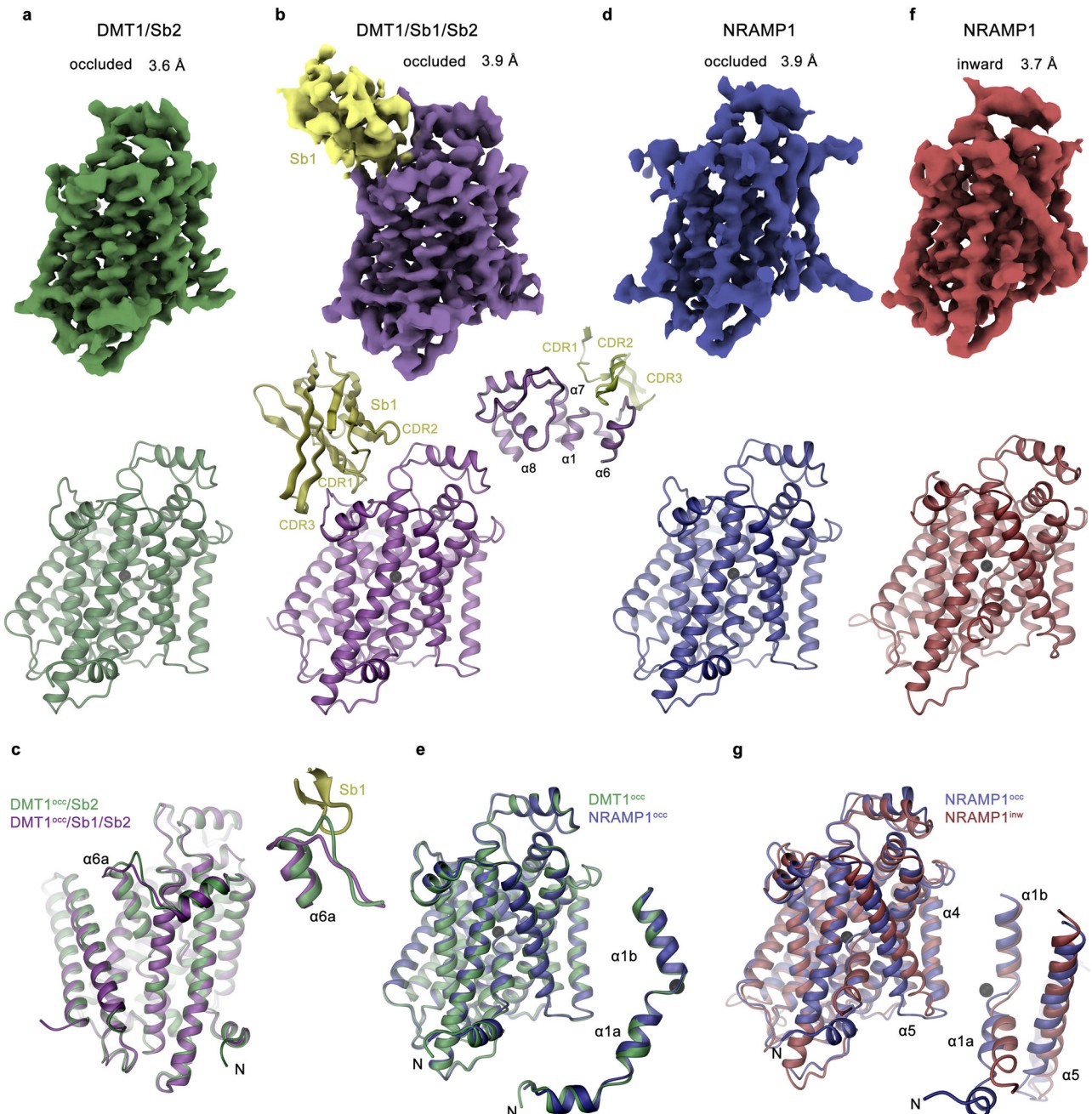

**Fig. 3 | Cryo-EM density and structural properties of human SLC11 paralogs. a,** **b** Cryo-EM density (top) and ribbon representation (bottom) of, **a** the DMT1/Sb2 complex at 3.6 Å and, **b** the DMT1/Sb1/Sb2 complex at 3.9 Å. In both cases no density of Sb2 is discernable. In **b**, Sb1 was well defined and is shown in yellow. Inset (right) shows blow-up of the interaction region between DMT1 and Sb1. **c** Superposition of the DMT1 structures from both datasets shows a similar occluded conformation of the protein. Small differences in the conformation of α6, which are induced by Sb1 binding are illustrated as inset on the top, right. **d** Cryo-EM density at 3.9 Å (top) and ribbon representation (bottom) of an occluded

conformation found in a population of particles in the dataset of NRAMP1. **e** Superposition of the occluded structure of NRAMP1 with the DMT1 structure illustrate the equivalence of both conformations. A blow-up of the superimposed α-helices 1 show the similarity in the conformation of both mobile elements. **f** Cryo-EM density at 3.7 Å (top) and ribbon representation (bottom) of an inward-facing conformation found in a population of particles in the dataset of NRAMP1. **g** Superposition of the inward-facing and occluded structures of NRAMP1 illustrate differences between both conformations. The largest differences are found in α-helices 1 and 5 shown as blow-up, right.

## Features of human SLC11 structures

The general architecture of both human SLC11 paralogs resembles the prokaryotic transporter EcoDMT (PDBID: 5M87, RMSD 2.0 Å), which contains 12 membrane-spanning α-helices, in contrast to the majority of prokaryotic homologs where the last α-helix is absent[21] (Supplementary Fig. 8). In the common protein fold of the family that was originally described for the amino acid transporter LeuT[31], the first five

α-helices form a unit that is topologically related to the following five α-helices, which are placed with opposite orientations in the membrane, and where unwound regions in the first helix of each repeat (i.e. α1 and α6) constitute a metal ion binding site located in the center of the protein (Fig. 4a, b)[20].

The correspondence of equivalent structures of the two human SLC11 paralogs to an occluded state is illustrated by the inaccessibility

**Table 1 | Cryo-EM data collection, refinement, and validation statistics**

| | DMT1-Sb2 (EMDB-50235) (PDB 9F6N) | DMT1-Sb1-Sb2 (EMDB-50236) (PDB 9F6O) | NRAMP1 inward-facing (EMDB-50237) (PDB 9F6P) | NRAMP1 occluded (EMDB-50238) (PDB 9F6Q) |
|---|---|---|---|---|
| Data collection and processing | | | | |
| Magnification | 130,000 | 130,000 | 130,000 | 130,000 |
| Voltage (kV) | 300 | 300 | 300 | 300 |
| Electron exposure (e–/Å2) | 59.9 | 60.47 | 60–63 | 60–63 |
| Defocus range (μm) | −2.4 to −1.0 | −2.4 to −1.0 | −2.4 to −1.0 | −2.4 to −1.0 |
| Pixel size (Å)* | 1.302 (0.651) | 1.302 (0.651) | 1.302 (0.651) | 1.302 (0.651) |
| Symmetry imposed | C1 | C1 | C1 | C1 |
| Initial particle images (no.) | 5,093,566 | 2,791,914 | 19,856,253 | 19,856,253 |
| Final particle images (no.) | 170,677 | 72,182 | 123,180 | 125,836 |
| Map resolution (Å) | 3.6 | 3.9 | 3.7 | 3.9 |
| FSC threshold | 0.143 | 0.143 | 0.143 | 0.143 |
| Map resolution range (Å) | 3.0–4.0 | 3.5–7.5 | 3.5–5.7 | 3.8–5.4 |
| Refinement | | | | |
| Initial model used (PDB code) | De-novo | 7P5V (Sb1) | P49279 AlphaFold | P49279 AlphaFold |
| Model resolution (Å) | 3.5 | 3.9 | 3.7 | 3.9 |
| FSC threshold | 0.143 | 0.143 | 0.143 | 0.143 |
| Model resolution range (Å) | | | | |
| Map sharpening B factor (Å2) | −60 | 0 | −100 | −50 |
| Model composition | | | | |
| Non-hydrogen atoms | 3652 | 4603 | 3720 | 3618 |
| Protein residues | 471 | 591 | 482 | 469 |
| Ligands | 1 | 0 | 1 | 0 |
| B factors (Å2) | | | | |
| Protein | 152.96 | 211.98 | 213.31 | 248.06 |
| Ligand | 150.98 | | 128.91 | |
| R.m.s. deviations | | | | |
| Bond lengths (Å) | 0.008 | 0.003 | 0.004 | 0.004 |
| Bond angles (°) | 1.195 | 0.635 | 0.949 | 0.826 |
| Validation | | | | |
| MolProbity score | 1.74 | 2.16 | 2.11 | 2.31 |
| Clashscore | 3.91 | 15.55 | 19.52 | 16.19 |
| Poor rotamers (%) | 2.80 | 2.24 | 1.79 | 2.62 |
| Ramachandran plot | | | | |
| Favored (%) | 96.59 | 96.76 | 97.29 | 95.93 |
| Allowed (%) | 3.41 | 3.24 | 2.71 | 4.07 |
| Disallowed (%) | 0.00 | 0.00 | 0.00 | 0.00 |

*Values in parentheses indicate the pixel size in super-resolution.

of the region harboring the conserved ion binding site from either side of the membrane (Fig. 4c). This assignment is also evident in the comparison to distinct structures of the prokaryotic DraNRAMP, for which several conformations on the transport cycle are known[22,23]. Upon superposition of the membrane-spanning helices, the RMSDs of 2.3 and 2.2 Å compared to the inward-occluded and outward-facing conformations of the bacterial transporter emphasize that the observed structure of DMT1 represents an intermediate state situated in-between the two conformations of the prokaryotic homolog (Fig. 4d–h, Supplementary Fig. 9a, b). The structural similarity between pro- and eukaryotic family members prevails despite the evolutionary distance between these two homologs that in the described case share an overall sequence identity of only 31% (Supplementary Fig. 8). With 568 amino acids, DMT1 is 163 residues longer than DraNRAMP. Extensions in DMT1 concern about 60 residues on the N-terminus, which, except for a small helix preceding α1a, are disordered in the

structure, more than 60 residues on the C-terminus, which form an additional 12th transmembrane segment that is not found in DraNRAMP and several insertions in loop regions (Fig. 4a, d, Supplementary Fig. 8). The most prominent extension in the extracellular region concerns an insertion between transmembrane segments 7 and 8 folding into two α-helices that are connected by a loop (Fig. 4d). Compared to the inward-occluded conformation DraNRAMP$^{occ}$, the highest structural correspondence is found in the core of the protein whereas differences in several peripheral regions are more pronounced (i.e. α-helices 5, 9 and 10) (Fig. 4e, Supplementary Fig. 9a, b). Compared to the outward-facing structure DraNRAMP$^{out}$, we find increased differences in the conformations of α-helices 1, 4 and 10, whereas the difference between α-helices 9 is decreased and the one between α-helices 5 has essentially vanished (Supplementary Fig. 9a, b). The intermediate position of DMT1$^{occ}$ compared to the two structures of DraNRAMP is also apparent in the location of α6a, which

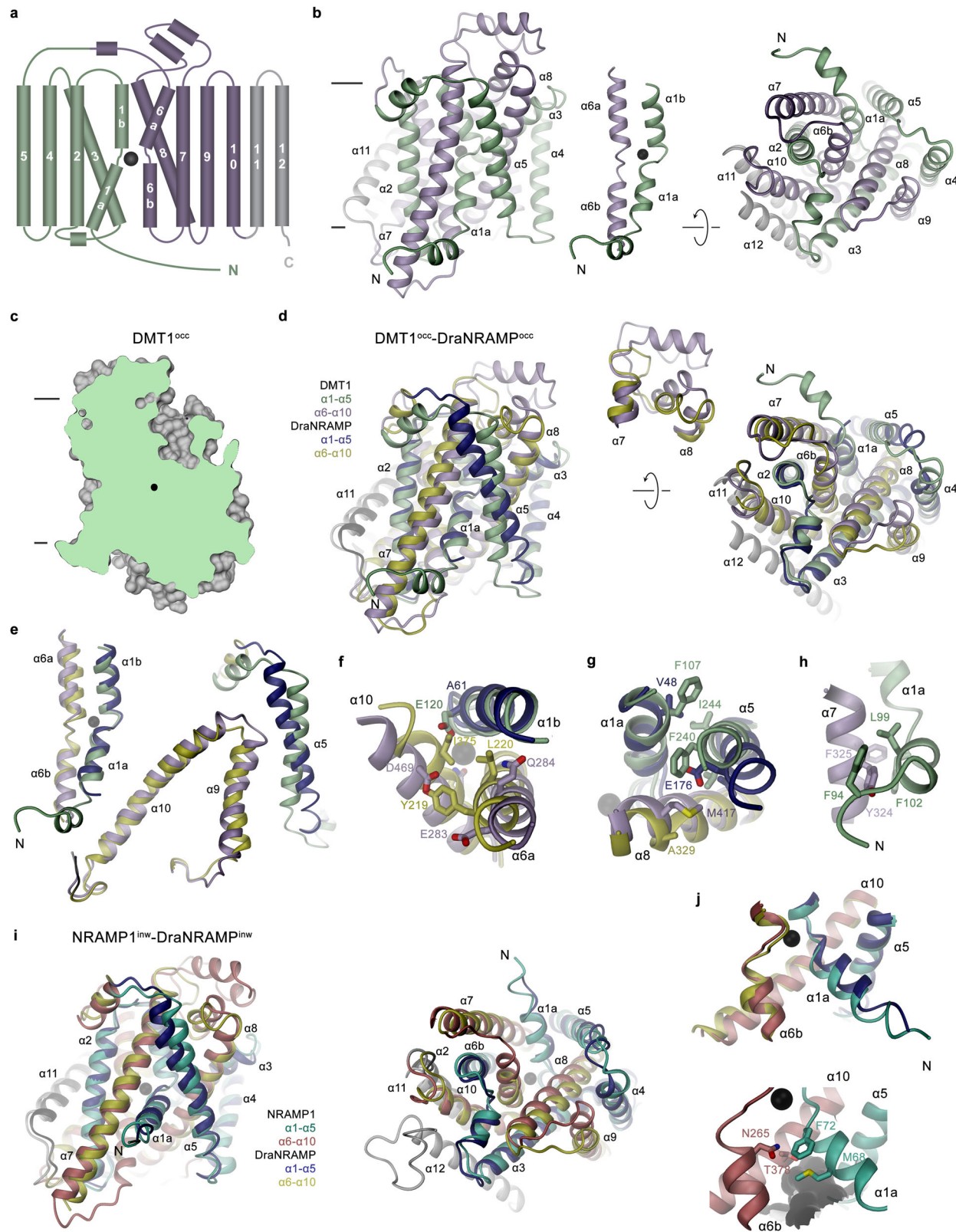

is placed in-between its inward-occluded and outward-facing conformations (Supplementary Fig 9b). A remarkable difference between DMT1 and DraNRAMP relates to the chemical properties of residues lining the aqueous cavity leading to the ion binding site in an outward-facing conformation. These are located at the interface between α-helices 1b, 6a, and 10, which together constitute the region blocking access to the metal ion binding site from an extracellular direction in

an occluded conformation. The three helices come closer in DraNRAMP$^{occ}$ than in DMT1$^{occ}$, where predominantly hydrophobic residues engage in tight interactions in the prokaryotic homolog whereas several substitutions with polar and charged residues render the region in the human transporters DMT1 and NRAMP1 more hydrophilic (Fig. 4f). A group of negatively charged residues in DMT1 (Glu 120, Glu 283 and Asp 469), would provide favorable electrostatics

**Fig. 4 | Structural relationships to prokaryotic transporters. a** Schematic topology of DMT1. The two structurally equivalent repeats of the protein are colored in green and violet, two additional α-helices at the C-terminus in grey. **b** Ribbon representation of DMT1$^{occ}$ viewed from within the membrane (left) and from the cytoplasm (right) with coloring as in **a**, the membrane boundary is indicated. Blow-up (center) shows the two partially unwound helices α1 and α6 constituting the metal ion binding site. **c** Slice through a surface of DMT1$^{occ}$, with the metal ion binding site indicated, illustrates the inaccessibility of the bound ion to either side of the membrane. **b**, **c** The membrane boundary is indicated. **d** Superposition of the DMT1$^{occ}$ structure and an occluded conformation of the prokaryotic homolog DraNRAMP (PDBID: 8E60, colored in blue, yellow and black). Inset (top center) shows the loop region, connecting α7 and α8, which is more extended in the human transporters. **e** View of selected helices from the superposition between DMT1$^{occ}$ and DraNRAMP$^{occ}$ shown in **d**. **f**, **g** View of selected helices occluding the access to the ion binding site from, **f** the extracellular side and, **g** the cytoplasm. Displayed are selected helices from the superposition shown in **d**. Side chains of residues lining the closed access path in both structures are shown as sticks. **h** Interactions with a short N-terminal helix preceding α1a with α7 in DMT1$^{occ}$. Sidechains of selected residues are shown as sticks. **i** Superposition of the NRAMP1$^{inw}$ structure (colored in cyan, red and grey) and the inward-facing conformation of the prokaryotic homolog DraNRAMP (PDBID: 8E6I, colored as in **d**). **d**, **i** Views are as in **b**. **j** Top, α-helices of the superposition shown in **i** lining the aqueous pocket leading from the cytoplasm to the binding site. Bottom, blow up of the same region in NRAMP1$^{inw}$ with selected residues blocking access to the ion binding site shown as sticks. The molecular surface of the aqueous cavity approaching the ion binding site from the cytoplasm is shown in grey. **a–g, i, j** A bound metal ion is indicated as black sphere.

for attracting divalent metal ions and could presumably be part of an inhibitory $Ca^{2+}$ binding site, which would stabilize the observed conformation and thus account for the described non-competitive inhibition of transport at high extracellular $Ca^{2+}$ concentrations[11,26] that was also observed in our experiments, although this assignment is at this stage hypothetical (Figs. 1f, 4f, Supplementary Fig. 1j). Conversely, on the closed intracellular access to the binding site, we find a tighter interaction in DMT1$^{occ}$ where α1a, the major mobile element upon the transition into an inward-facing state, engages in tight interactions with α5 and α8, which are much less pronounced in the prokaryotic homolog (Fig. 4g). In this region, we also find interactions between the short α-helix preceding α1a and α7, which might further stabilize the observed conformation (Fig. 4h). However, it is unclear whether these differences are characteristics of the human transporters or whether the observed conformation represents an intermediate state that has further progressed in the transition towards an outward-facing conformation.

Whereas one of the structures obtained for NRAMP1 (NRAMP1$^{occ}$) closely resembles the occluded conformation obtained for its paralog DMT1 (Fig. 3d, e), a second population of particles reside in a distinct conformation that is distinguished by an outward movement of α-helix 1a and concomitant changes of α4 and 5 leading to the dissociation of tight contacts between α1a, α5 and α8 and the consequent opening of an intracellular aqueous path towards the binding site (Fig. 3f, g). This structure (NRAMP1$^{inw}$) resembles the inward-facing conformation of DraNRAMP (DraNRAMP$^{inw}$), where the prokaryotic protein has undergone similar, although slightly more extended changes, compared to its occluded state, particularly with respect to α1a and α5 (Fig. 4i, j, Supplementary Fig. 9c, d). In the case of DraNRAMP$^{inw}$ the described movement has opened an aqueous access path to the metal ion binding site located in the center of the protein (Fig. 4i, j). Although a similar aqueous cavity is also found in NRAMP1$^{inw}$, as a consequence of the smaller extent of the changes, the access to the actual binding site appears still blocked by protein residues (*i.e.* Phe 72, Asn 265, and Met 68) and the respective structure does thus probably not present a full transition into an inward-facing conformation (Fig. 4j).

## Features of the ion binding site

As previously defined in the structures of prokaryotic homologs[20,23], each SLC11 transporter contains a metal ion binding site that is located in the center of the protein at the unwound regions of α-helices 1 and 6 (Fig. 4b). Both helices contribute conserved residues that participate in metal ion binding. These include the side-chains of an aspartate (DMT1: Asp 115) and an asparagine (DMT1: Asn 118) on α1, as well as the backbone carbonyl of an alanine (DMT1: Ala 291) on α6 all providing hard ligands for ion coordination (Fig. 5a). Additionally, a conserved methionine on a6 (DMT1: Met 294) binds the transported transition metal ion via coordinative interactions with the free electron pairs of the thioether group, thus providing a soft ligand that would not form similarly strong interactions with alkaline earth metal ions (Fig. 5a).

While metal ion interactions prevail in all conformations, they are supposed to be most extended in an occluded state of the transporter where the ion is largely surrounded by either protein residues or single trapped water molecules, and where the access to this site is sealed from both sides of the membrane (Fig. 4c). This state is adopted in the DMT1$^{occ}$ structure obtained in presence of $Mn^{2+}$, where residual density at the binding site presumably corresponds to a bound transition metal ion (Fig. 5b, c). This ion is in interaction distance with the aforementioned residues and with the sidechain of Gln 475 located on α-helix 10 (Fig. 5a). Besides the described close contacts, the same region also harbors residues at somewhat larger distance including Ser 189 and Gln 192, both on α3, which potentially also contribute to ion stabilization by either providing weaker direct or water-mediated interactions in a second coordination shell (Fig. 5d). The comparison with the corresponding occluded conformation of the prokaryotic DraNRAMP shows general resemblance but also certain differences. In DraNRAMP$^{occ}$, the ion was shown to be surrounded by protein residues and one trapped water molecule, which together contribute to its preferred octahedral coordination[23] (Fig. 5e). Similarities and differences in ion interactions between DraNRAMP$^{occ}$ and DMT1$^{occ}$ are best illustrated in a local superposition of the regions constituting the bulk of the binding site composed of the unwound parts of α1 and α6 and residues on α10. While this superposition leads to an overlap of the ion coordinating sidechains on α1 and α6, it also illustrates the disparity in the location of the bound metal ions in both structures, which are separated by 3.6 Å (Fig. 5f). In DraNRAMP, the ion positioned in the center of the unwound part of both helices is tightly surrounded by the corresponding residues on α1 and α6. Besides the described sidechain coordination, interactions include contacts with the backbone amide oxygen of Ala 227 on α6a and its symmetry mate Ala 53 on α1a (Fig. 5e). The only additional interaction is mediated by a bound water molecule, bridging the metal ion to the sidechain of Gln 378 on α10, whose comparably poor superposition in both structures indicates differences between DraNRAMP and DMT1. The presumed location of the ion in DMT1 is shifted towards α10 to undergo direct interactions with Gln 475 (Fig. 5a, d, f). In the sequence alignment of DraNRAMP and DMT1, this residue is not at the equivalent position of Gln 378 in DraNRAMP but instead is displaced by one helix turn towards the cytoplasm at a position which is occupied by a leucine (Leu 381) in the prokaryotic transporter (Supplementary Figs. 8 and 10a). Potentially as a consequence of their lower resolution, we do not find indication of bound ions in the maps of the DMT1/Sb1/Sb2 complex and the occluded conformation of NRAMP1, whereas residual density at the site in the inward-facing structure of NRAMP1 indicates a position that resembles the equivalent location in the inward-facing structures of the prokaryotic ScaDMT[20] and DraNRAMP[23] (Fig. 5g, Supplementary Fig. 10b).

Besides the ion binding site, we also find structural features of presumed $H^{+}$ pathways conserved between the pro- and eukaryotic transporters which branch from the binding site in intracellular

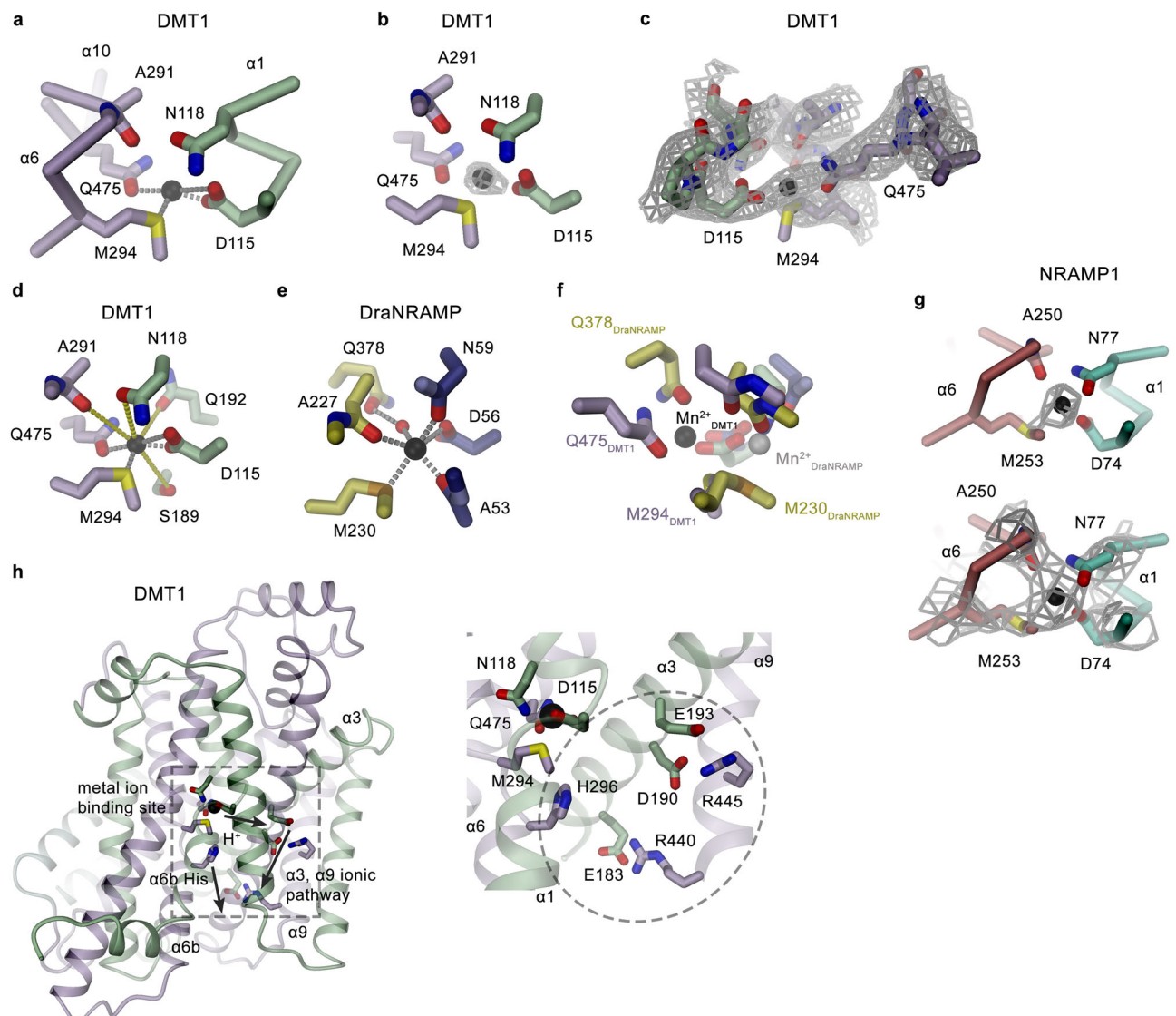

**Fig. 5 | Metal ion binding site. a** Structure of the metal ion binding site in DMT1. Selected residues surrounding the bound $Mn^{2+}$ (black) are shown as sticks. **b** Residual density in the map obtained from the DMT1/Sb2 dataset at 3.6 Å (contoured at 13 σ) assigned to a $Mn^{2+}$ ion (black sphere) with surrounding residues displayed as sticks. **c** Cryo-EM density of the same map (contoured at 8 σ) superimposed on residues of the metal ion binding site. The view is rotated by 180° compared to **b**. **d** Presumable metal ion interactions in DMT1$^{occ}$. **e** Metal ion interaction in the prokaryotic DraNRMP$^{occ}$ structure showing an inward-occluded state (PDBID: 8E60). **a, d, e** Presumable protein-metal ion interactions (distance <4 Å) are indicated by dashed grey lines, in **d**, distances ranging between 4 and 5 Å are shown as yellow dashes lines. **f** Superposition of the ion binding sites of DMT1$^{occ}$ and

DraNRAMP$^{occ}$. The positions of the respective bound ions are indicated as spheres (black, DMT1, grey DraNRAMP). **g** Residual density in the NRAMP1$^{inw}$ map at 3.7 Å (contoured at 18 σ) assigned to a $Mn^{2+}$ ion (black sphere) (top) and section of the same map (contoured at 13 σ) surrounding the metal ion binding site (bottom). The backbone is displayed as Cα trace with surrounding residues shown as sticks. **h** Residues in DMT1 whose equivalents in several SLC11 homologs were implied to play a role in H$^+$ transport including a conserved His on α6 and an ionic interaction network on α3 and α9, in relation to the metal binding site. The protein is shown as a ribbon (left). The dashed rectangle marks the region of interest displayed as a blow-up (right). Residues of the putative H$^+$ efflux pathway are placed within the dashed circle (right).

direction (Fig. 5h). The residues include a conserved histidine (DMT1: His 296) located on α6b intracellular to the ion coordinating methionine (Fig. 5h, Supplementary Fig. 10c) and three conserved acidic residues on α3, which reside on equivalent positions as found in all prokaryotic homologs functioning as H$^+$ coupled metal ion transporters (DMT1: Glu 183, Asp 190, Glu 193) (Fig. 5h, Supplementary Figs. 8 and 10d). These acidic residues are in interaction distance to two arginines located on α9 (DMT1: Arg 440, Arg 445), which, due to the poor superposition of this helix compared to the prokaryotic transporters originate from different positions of the helix while their sidechains undergo equivalent interactions to form an extended ionic network (Fig. 5h, Supplementary Figs. 8 and 10d). These residues were

previously assigned a role in proton release to the intracellular side with mutations compromising H$^+$ transport, although the detailed mechanism of proton coupling in the SLC11 family is still debated[21,32].

## Functional properties of ion binding site mutants

After the identification of residues that coordinate the transported transition metal ions, we have constructed point mutants in these positions and investigated their effect on $Mn^{2+}$ and $Fe^{2+}$ transport in DMT1. In case of the two residues on α1, whose sidechains contribute hard ligands for ion coordination, the corresponding replacement to Ala in the mutants D115A and N118A strongly diminished transport of both transition metal ions (Fig. 6a-c, Supplementary Fig. 11a–f).

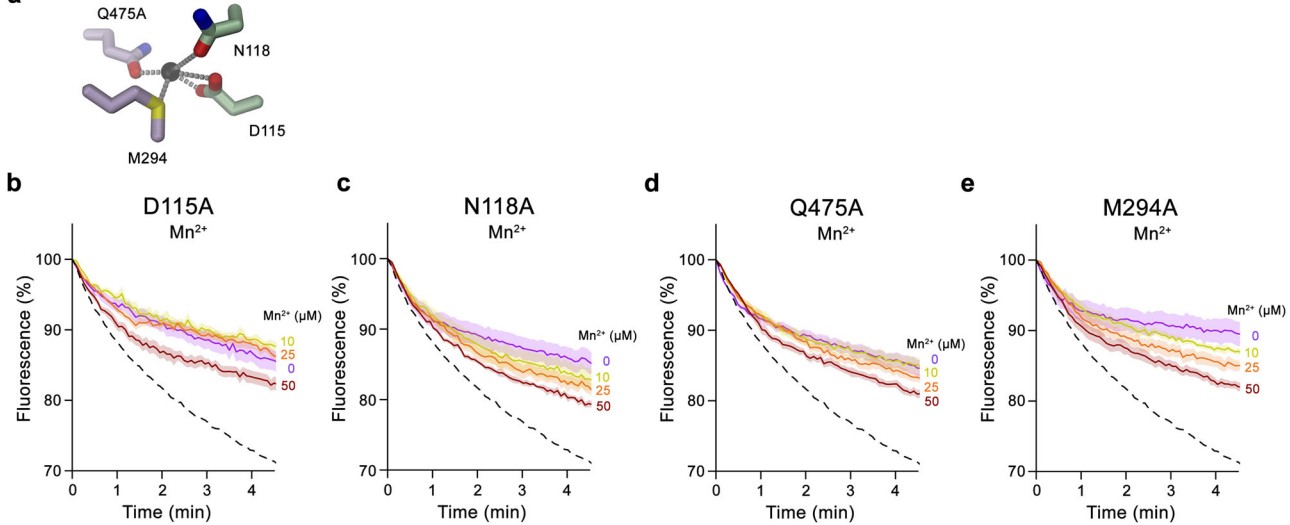

**Fig. 6 | Transport properties of metal binding site mutants. a** Transition metal ion binding site of DMT1 with side-chains of mutated residues shown. Metal ion interactions are represented by dashed lines. **b–d** Mn²⁺ transport into proteoliposomes containing metal binding site mutants of DMT1: **b**, D115A, **c**, N118A, **d**, Q475A, **e**, M294A. **b–d** Time and concentration dependence uptake assayed by the quenching of the fluorophore calcein trapped inside the vesicles. Data show mean of eight experiments from two independent reconstitutions, errors are s.e.m. Fluorescence is normalized to the value after addition of substrate (t = 0). Applied ion concentrations are indicated. Trace of WT upon addition of 50 μM Mn²⁺ is shown as a dashed line for comparison.

Similarly, the mutation of Gln 475 located on α10 to Ala strongly impairs Mn²⁺ and Fe²⁺ transport, consistent with its presumed role in ion interactions observed in the DMT1 structure (Fig. 6d, Supplementary Fig. 11g–i). The transport phenotypes resemble equivalent effects in prokaryotic transporters[20,21,33], including the mutation of Gln 378 in DraNRAMP[22,33], which is the functional equivalent of Gln 475 in DMT1 located one helix turn in extracellular direction (Supplementary Figs. 8 and 11a). Finally, we probed the mutation of Met 294 on α6 providing a soft ligand for transition metal ion coordination and found a similar deleterious effect on Mn²⁺ and Fe²⁺ transport as for the mutations of residues on α1 and α10 (Fig. 6e, Supplementary Fig. 11j–m). This pronounced phenotype is in contrast to a more moderate effect of the equivalent mutations in the prokaryotic transporters EcoDMT and DraNRAMP, where robust Mn²⁺ transport in the mutant proceeds with a somewhat higher $K_m$[34,35]. In both prokaryotic homologs, these mutants also showed Ca²⁺ transport activity[34,35], which appears not to be the case in human DMT1 as reflected in the non-detectable signal in the Fura-2 assay (Supplementary Fig. 11n). Thus, it appears that the human transporter has evolved a stricter selectivity for transition metals than its prokaryotic counterparts.

## Discussion

By elucidating the architecture and functional properties of DMT1 and NRAMP1, the two human members of the SLC11 family, our study has defined the structural framework for iron transport in the context of human metabolism and pathogen resistance. Additionally, it has provided a direct characterization of the transport properties of NRAMP1, whose intracellular localization has precluded detailed functional investigations (Fig. 7a). Experiments using reconstituted protein show that both human SLC11 paralogs share the ability to transport the transition metal ions Fe²⁺ and Mn²⁺ and the toxic metal ion Cd²⁺, while efficiently discriminating against the alkaline earth metal ions Ca²⁺ and Mg²⁺ (Fig. 1, Supplementary Fig. 1d–f). These findings conform with previous studies of DMT1 that were based on cellular uptake and electrophysiology[4,36] and a study that has assayed NRAMP1 mediated metal uptake into cells from overexpressed protein targeted to the plasma membrane[13]. A difference between both transporters concerns a weak and presumably non-competitive inhibition of DMT1 at low mM

Ca²⁺ concentrations that we did not detect in case of NRAMP1 (Fig. 1f, Supplementary Fig. 1j, k). In this respect it is remarkable to find acidic residues at the extracellular entrance to the binding site of DMT1, whose equivalent positions are occupied by hydrophobic residues in prokaryotic transporters and which would be candidates for a weak Ca²⁺ binding site (Figs. 4, 7b). This site would be accessible from the extracellular side with Ca²⁺ binding stabilizing DMT1 in a conformation that precludes the access of transition metal ions. Remarkably, one of these positions located on α6a (DMT1: Glu 283) is replaced by leucine in NRAMP1 (Leu 242) which would disfavor a similar interaction (Supplementary Fig. 8). However, this proposal is at this stage hypothetical and still requires experimental validation. The inability of both proteins to interact with Mg²⁺ (Fig. 1g, h) excludes this ion as transported substrate and thus refutes the proposal that NRAMP1 would be directly involved in the deprivation of engulfed microbes from Mg²⁺ (ref. 17). With an unusually small ionic radius (of 0.65 Å) and a consequent strongly increased surface charge density, the dehydration of this divalent metal ion is energetically costly. The transport of Mg²⁺ thus requires distinct metal ion binding site properties, which permit interactions with an ion that has retained part of its hydration shell, as recently shown for a prokaryotic branch of the family termed NRAMP related Mg²⁺ transporters (NRMT)[34].

As in their prokaryotic counterparts, metal ion transport in both human paralogs is coupled to the symport of protons (Figs. 2 and 7a), which clarifies a controversy on the transport mechanism of NRAMP1 (refs. 13–16.). As previously described for DMT1 (ref. 12.), our studies have also revealed an uncoupled H⁺ leak in absence of transported metal ions that is even more pronounced in NRAMP1 (Fig. 2, Supplementary Fig. 2d). The uncoupled H⁺ leak can be suppressed by the inhibitor TMBIT (Fig. 2d, e, Supplementary Fig. 2f), which was previously characterized to compete with metal ion binding in pro- and eukaryotic SLC11 transporters and stabilize the protein in an outward-facing conformation[28], suggesting that H⁺ cannot access the proton pathway in this inhibited state (Fig. 7b).

The comparison of the structures of both mammalian transporters with corresponding conformations of prokaryotic homologs reveals congruent features but also differences that are a consequence of the divergence of the proteins in terms of their size and sequence.

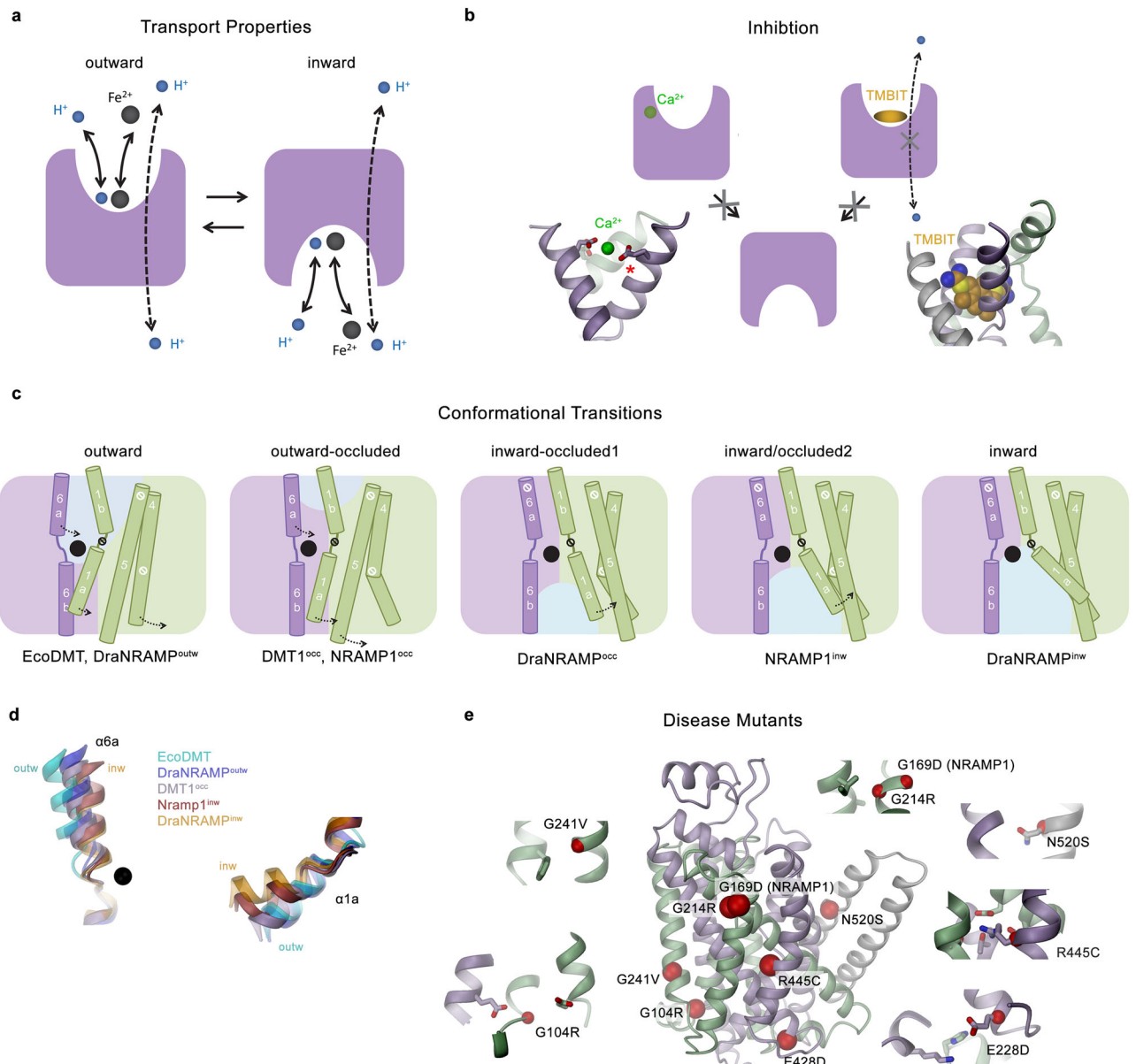

**Fig. 7 | Mechanistic properties of human SLC11 transporters. a** Schematic depiction of an SLC11 protein catalyzing transition metal ion transport that is coupled to the symport of H$^+$ by an alternate access mechanism in parallel to an uncoupled H$^+$ leak. **b** Scheme of the inhibition of the transporter by Ca$^{2+}$ and the small molecule TMBIT, both acting from the extracellular side. Left, Ca$^{2+}$ inhibition. This weak and non-competitive inhibition is found in DMT1 and not shared by its paralog NRAMP1. Model of a possible inhibitory Ca$^{2+}$ binding site composed of acidic residues. The glutamate on α6a (marked by red asterisk) is specific to DMT1 and replaced by a leucine in NRAMP1. Right, inhibition by the benzylisothiourea derivative TMBIT, which binds to the aqueous pocket leading to the binding site in an outward-facing conformation thereby inhibiting coupled and uncoupled transport. The binding position is inferred from structures of the prokaryotic

EcoDMT (PDB: 6TL2). **c** Depiction of structural changes relating the known conformations of SLC11 transporters including the conformations of the two human family members determined in this study. **d** Conformational differences of helices α6a and α1a acting as primary gating elements that regulate the access to the metal ion binding site in different structures of SLC11 transporters, illustrating the intermediate conformations of human transporters between outward- and inward-facing states. **e** Location of mutations in DMT1 and NRAMP1 that were associated with diseases. Model (center) shows a ribbon representation of DMT1 with mutated positions highlighted by red spheres. Insets, show blow-ups of the respective regions with selected side-chains shown as sticks and mutated positions additionally labeled by a red sphere.

Besides the presence of additional structural elements at both termini and in extended loop regions, these distinctions are manifested in the relative orientation of transmembrane helices, which, although more pronounced in peripheral regions, extend to differences even in vicinity of the binding site (Figs. 4 and 5, Supplementary Figs. 9 and 10). With respect to their conformations, the two human paralogs presumably display intermediates on the transport cycle, which in case of the occluded structures of DMT1 and NRAMP1 would be placed between outward-open and inward-occluded conformations found in

the prokaryotic transporter DraNRAMP, where the intracellular gate precluding access to the metal binding site has closed while the extracellular gate has not yet opened (Fig. 7c, Supplementary Fig. 9a, b). This intermediate position is manifested in differences between α5 and α9 compared to the inward-occluded state of DraNRAMP and the conformation of α6a, which constitutes the principal element that controls gate opening to the extracellular side (Fig. 7c, d, Supplementary Fig. 9a, b). The orientation of α6a is positioned between the inward-occluded and the outward-facing

conformation of DraNRAMP[22,23] and is even more pronounced in case of the outward-facing conformation of EcoDMT[21] (Fig. 7c, d). In the inward-facing structure of NRAMP1, we find a conformation where α1a has moved to open an intracellular access path to the binding site but where the opening is still incomplete as judged from the smaller amplitude of the movement in NRAMP1 compared to the prokaryotic transporter and the consequently still restricted access to the binding site (Figs. 3g, 4i, j and 7c, d). Analogously, with respect to the interaction with transported metal ions, we find general correspondence but also differences in DMT1 compared to prokaryotic homologs. Assuming that residual densities at the binding sites of DMT1$^{occ}$ and NRAMP1$^{inw}$ correspond to bound metal ions that were present in the sample at low millimolar concentrations (Fig. 5b, c, and g), we find a similar coordination in the inward-facing structure of NRAMP1 compared to the prokaryotic homologs DraNRAMP and ScaDMT (Supplementary Fig. 10b). In contrast, the presumable position of the ion in DMT1$^{occ}$ is shifted by about 3.6 Å compared to the position in DraNRAMP$^{occ}$, allowing similar ion interactions as found in prokaryotic transporters but additionally also a specific coordination with a residue located on α10 (i.e. DMT1: Gln 475) (Fig. 5a, d–f). A comparable interaction on DraNRAMP with a glutamine located one helix turn apart in extracellular direction (DraNRAMP: Gln 378) was found to be mediated by a bound water molecule[23] (Fig. 5e, Supplementary Fig. 10a). With respect to functional properties of mutants of the ion binding site, we find similar consequences as for equivalent positions in prokaryotic transporters with mutations of residues providing hard ligands located on α1 and α10 essentially abolishing metal ion transport (Fig. 6b–d). A different phenotype was observed for a mutation of a conserved methionine on α6 (DMT1: Met 294), which contributes a soft ligand for the interaction with transition metal ions in the WT protein. In prokaryotic transporters, the mutation of the equivalent residue to alanine yields a protein that has retained $Mn^{2+}$ transport and was also able to transport $Ca^{2+}$, leading to the proposal that the main role of this methionine concerns the counter-selection against alkaline earth metal ions[34,35]. In contrast, the equivalent mutation M294A in DMT1 has a different effect as neither transport of $Mn^{2+}$ nor $Ca^{2+}$ was observed, pointing towards a stronger selectivity for transition metal ions in human transporters (Fig. 6e, Supplementary Fig. 11m, n).

Besides providing insight into transport mechanisms, our study permits the localization of residues whose mutation in DMT1 and NRAMP1 were associated with iron storage-related disorders and a decreased resistance against bacterial infections (Fig. 7e). In DMT1, two of these mutations (i.e. G241V and R445C) result in compromised splicing of mRNA, leading to a loss of function as consequence of a premature termination of translation[37–40]. In contrast, the mutation G104R at the beginning of α1a results in a folded protein, whose equivalent in DraNRAMP was shown to disrupt transport by stabilizing an inward-open conformation[22,41] (Fig. 7e). Similarly, the mutant E428D located at the intracellular end of α8 (Fig. 7e) was described to fold properly but resulted in considerably lower expression levels[39]. Other mutations, such as G214R located at α4 at the interaction interface with α3, the α9 residue R445C, which engages in salt bridge interactions with residues in α3 as part of a putative H$^+$ pathway, and N520S on α11 in the interface with α10 lead to compromised folding and mislocalization[37,40,42,43] (Fig. 7e). In case of NRAMP1, the mutation D543N, which is located after the last helix α12 at the unstructured C-terminus increases the susceptibility to tuberculosis[44,45] (Fig. 7e). Finally, the mutant G169D (i.e. Gly 172 in the here used construct), which is located in the same region as Gly 214 in DMT1, was identified in mice to reduce the resistance against intracellular pathogens and to lead to iron overload in macrophages in the spleen and liver[8,46].

Collectively, the data provided in this study provides a detailed structural and functional framework for the understanding of a central metabolic process related to the acquisition of the most abundant trace element in our body and thus provides a basis for the development of specific modulators interfering with their function, which constitutes a strategy for the treatment of iron overload disorders[47].

## Methods

### Cloning and DNA preparation

The codon-optimized genes for mammalian expression of the human DMT1 (UniProt identifier P49281-3) and NRAMP1 (UniProt identifier P49279-1) were synthesized by GeneScript and cloned via FX-cloning[48] into the expression vector pcDXC3MS. This vector encodes for a C-terminal HRV-3C cleavage site followed by a Myc- and a Streptavidin-Binding Peptide-tag. Point mutants were generated by site-directed mutagenesis[49], using Phusion polymerase and primers specified in Supplementary Table 3. All constructs were transformed into *E. coli* MC1061 cells and cultured in TB media at 37 °C overnight. Cultures were harvested by centrifugation at 4500 x *g* for 20 min. DNA was purified using a NucleoBond PC10000 kit according to the manufacturer's manual. Purified plasmids were resuspended in sterile water to 1 mg ml$^{-1}$ concentration and stored at −20 °C until further use.

### Expression and purification of DMT1 and NRAMP1

Suspension HEK293S GlnTI$^-$ cells (ATCC: CRL-3022) were grown in HyClone Trans FY-H medium (supplemented with 1% fetal bovine serum, 4 mM L-glutamine, 100 U ml$^{-1}$ penicillin/streptomycin and 1.5 g l$^{-1}$ kolliphor-P188) at 37 °C and 5% $CO_2$ content and shaking at 180 rpm. Cells were seeded at $0.5 \times 10^6$ cells ml$^{-1}$ 24 h prior to transient transfection. To transfect, 1.3 μg plasmid for 1 million cells was diluted in DMEM to 8 μg ml$^{-1}$ and PEI was added to a final concentration of 20 μg ml$^{-1}$. After 30 min of incubation at room temperature, 50 ml of this transfection mixture was added to 300 ml suspension culture, at a cell density of $10^6$ cells/ml, then 5 mM valproic acid was added to the cells. Transfected cells were harvested after 36 h by 15 min centrifugation at 550 x *g* at 4 °C. Pellets were washed with ice-cold PBS, collected by centrifugation at 1000 x *g* for 10 min at 4 °C, flash frozen in liquid nitrogen and stored at −80 °C until further use.

All purification steps were carried out at 4 °C. Cells were lysed in lysis buffer (50 mM HEPES at pH 7.4, 150 mM NaCl, 2% DDM, 0.2% CHS, 2 mM $MgCl_2$, protease inhibitors, and DNaseI) by incubation for 30 min under constant stirring. Lysate was cleared by centrifugation at 15,000 x *g* for 20 min. StrepTactin Superflow affinity resin was added to the cleared supernatant and incubated for 30 min under gentle agitation. The suspension was loaded onto a gravity flow column. The resin was washed once with 10 column volumes (CV) of lysis buffer, four times with 10 CV of wash buffer (50 mM HEPES at pH 7.4, 150 mM NaCl, 0.04% DDM, and 0.004% CHS) and eluted with 5 CV of elution buffer (50 mM HEPES at pH 7.4, 150 mM NaCl, 0.04% DDM, 0.004% CHS and 5 mM d-desthiobiotin). The eluate was concentrated using 50 kDa MWCO centrifugal filters. To remove the fusion tag, HRV 3C protease was added to the protein at a molar ratio of 1:5 (protease:protein), and incubated for 1 h on a rotating wheel. The sample was centrifuged at 10,000 x *g* for 10 min to remove precipitate and subjected to size exclusion chromatography on a Superdex 200 10/300 GL column equilibrated with SEC buffer (10 mM HEPES at pH 7.4, 150 mM NaCl, 0.03% DDM and 0.003% CHS). For the sample that was used for structure determination, CHS was omitted from the SEC buffer. Peak fractions were collected and concentrated using 50 kDa MWCO centrifugal filters.

### Synthetic nanobody selection, expression and purification

The selection against DMT1 was performed as described[29] using 0.04% DDM and 0.004% CHS when detergent was needed. Protein biotinylation was carried out using the EZ-Link NHS-PEG4-Biotinylation kit (Thermo Fisher Scientific) according to the manufacturer's instructions. DMT1 has neither been frozen nor stored overnight and was always used freshly for the selection. The mRNA libraries and vectors

needed for the selection were generously provided by Prof. Markus Seeger (Institute of Medical Microbiology, UZH). The selected sybodies were subcloned into pSBInit vectors. These constructs were expressed and purified as described[29], except that EDTA was not added to the SEC buffer. The purified sybodies were concentrated to 4–12 mg ml⁻¹, flash frozen, and stored at −80 °C until further use.

### SPR binding assay

Chemically biotinylated DMT1 or NRAMP1 (EZ-Link NHS-PEG4-Biotinylation kit) was immobilized on an equilibrated (10 mM HEPES pH 7.4, 150 mM NaCl, 0.04% DDM, 0.004% CHS and 0.1% BSA) SAD200M sensor chip at a density of around 900 RU. Flow cell 1 was left empty and used as a reference cell. The system was washed for 2 h with running buffer (10 mM HEPES pH 7.4, 150 mM NaCl, 0.04% DDM, 0.004% CHS, and 0.1% BSA). Analytes were injected at 10 °C at a flow rate of 30 µl min⁻¹ at 7.8, 15.6, 31.25, 62.5, 125, 250 and 500 nM concentrations. Experiments were carried out on a BiaCore T200 machine (Cytiva Life Sciences) and data was fitted to a single-site binding model with the BIAevaluation software v. 4.1 (GE Healthcare). To test whether DMT1 binds both sybodies simultaneously, Sb1 and Sb2 were co-injected at a concentration of 2 µM each, and the resulting signal was compared to the individual application of each sybody at concentrations of 2 and 4 µM at a flow rate of 30 µM min⁻¹ at 10 °C.

### Cryo-EM sample preparation and data collection

For cryo-EM sample preparation the purified DMT1 (at a final concentration of 2.3 mg ml⁻¹) was mixed with purified sybodies (at final concentrations of 1.1 mg ml⁻¹). NRAMP1 was used at a concentration of 3.6 mg ml⁻¹. MnCl₂ was added to all samples at a final concentration of 5 mM. 2.5 µl of each protein sample was applied to a glow discharged grid (Quantifoil, holey carbon grids, R 1.2/1.3, AU 300 mesh), blotted for 2-4 s and plunge frozen in liquid ethane-propane using a vitrobot (Thermo Fischer, Mark IV) in a controlled environment (4 °C, 100% humidity). Grids were clipped and stored in liquid nitrogen.

Datasets were collected on a 300 kV Titan Krios G3i microscope (Thermo Fischer Scientific), equipped with a K3 direct electron detector (Gatan) operating in super-resolution mode, a post-column BioQuantum energy filter (Gatan) with 20 eV slit and with an objective aperture of 100 µm. Micrographs were acquired at 130,000× magnification, at varying defocus ranging from −1 to −2.4 µm, an exposure of 1.21 seconds (47 frames), and an approximate dose of 1.8 e⁻/Å²/frame using EPU. Micrographs were collected with a pixel size of 0.651 Å (0.3225 Å in super resolution) and binned two-fold by EPU (v 2.9). The total electron dose for all datasets was in the range of 62 e⁻/Å².

### Data processing

All datasets were processed in a similar manner using cryoSPARC[50] (v. 3.1-v. 4.5). Micrographs were subjected to patch motion correction followed by patch CTF estimation. High quality micrographs were selected based on ice thickness, CTF estimation and motion trajectories resulting in a total number of 6,564 micrographs for the DMT1/Sb1/Sb2 complex, 19,810 micrographs for the DMT1/Sb2 complex and 57,783 micrographs for NRAMP1 (Supplementary Table 3, Supplementary Figs. 4–6). Circular and elliptical blobs ranging between 120-180 Å were picked from 10 Å low-pass filtered micrographs. High quality particles were selected based on their power spectra and NCC scores and extracted and binned 4-fold to speed up processing. The extracted particles were subjected to several rounds of 2D classification. As no clear protein features were discernable in the 2D averages, promising classes were selected based on the dimensions and shape of the averaged particles, and those where at least one sybody was visible. Multiple volumes were generated using ab-initio modeling with initial and final resolutions of 12 and 7 Å respectively, and with the number of initial and final iterations set to 1,000. The most promising volumes were selected and subjected to further ab-initio 3D classification with

the described settings. Once a 3D volume with discernable trans-membrane helices was obtained, multiple rounds of heterogenous refinements were carried out using one high and multiple low-quality volumes which were previously generated. When the estimated resolution was approaching the Nyquist resolution of 5.3 Å, the selected particles were re-extracted with a box size of 384 pixels, Fourier cropped to 192 pixels, and subjected to further heterogenous refinement. The best particles were subjected to ab-initio modeling with two classes with high class similarity and initial and final resolutions of 7 and 4 Å, respectively. The best particles were subjected to non-uniform refinement using the highest quality initial model, and the resulting 3D volume was used to generate 2D templates for template picking on 7 Å low-pass filtered micrographs. The selected particles were subjected to the same processing pipeline as described above. The final map was obtained from local refinement with a mask excluding the detergent micelle. In cases where multiple datasets were collected, the new micrographs were directly subjected to template picking, and merged during heterogenous refinement.

### Model building and refinement

The model of DMT1/Sb2 complex was built de-novo in Coot[51] (v. 0.9.8.91). For the DMT1/Sb1/Sb2 ternary complex, the previously built model for the DMT1/Sb2 complex and a model for a sybody (PDB: 7P5V) was used as reference. The models for NRAMP1 were built using the AlphaFold2 model P49279 (https://alphafold.ebi.ac.uk/). All models were refined using PHENIX[52] (v. 1.21) with iterative manual adjustments in Coot. Figures displaying molecular structures were prepared with DINO and figures displaying densities with ChimeraX[53] (v. 1.3).

### Proteoliposome reconstitution

Chloroform solubilized POPE, POPG, and cholesterol were mixed at a molar ratio of 7:2:1 in a glass vial. Lipids were dried under a gentle flow of nitrogen stream at room temperature while rotating, forming a thin lipid film on the glass wall. The dried lipids were washed once with diethyl ether and dried in the same manner. The dried lipids were placed under vacuum for 2 h to allow the residual solvent to evaporate. The dry lipid film was resuspended in liposome buffer (5 mM HEPES pH 7.4, 100 mM KCl) under gentle shaking to 20 mg ml⁻¹. The hydrated liposomes were pooled and briefly sonicated for 30 s, aliquoted, and subjected to 2 freeze-thaw cycles. The liposomes were then flash-frozen in liquid nitrogen and stored at −80 °C until further use.

On the day of reconstitution, liposomes were thawed at room temperature and extruded 13 times through polycarbonate membranes with a pore size of 400 nm (Avestin) and diluted to 4 mg ml⁻¹ in a 100 mM KCl solution. Liposomes were destabilized as described[54] and protein was added to the liposomes at a protein to lipid ration of 1:100 (w/w). In case of empty liposomes (control), an equivalent volume of SEC buffer was added. Detergent was removed as described previously[54], liposomes were harvested by ultracentrifugation at 236,000 x g for 30 min, resuspended in proteoliposome buffer (1 mM HEPES pH 7.4, 100 mM KCl) to 40 mg ml⁻¹ lipid concentration, 50 µl aliquots were flash frozen in liquid nitrogen and stored at −80 °C until further use.

### Fluorescence-based transport assays

To assay divalent metal transport, one liposome aliquot was diluted with inside buffer (20 mM HEPES pH 7.4, 100 mM KCl and either 250 µM Calcein (Invitrogen) or 100 µM Fura-2 (Thermo Fischer Scientific)) to a final volume of 450 µl. The liposomes were subjected to 3 freeze-thaw cycles and extruded 13 times through a polycarbonate membrane with a pore size of 400 nm (Avestin). Liposomes were collected 3 times by centrifugation at 170,000 x g for 20 min at room temperature, washed twice with wash buffer (20 mM HEPES pH 7.4, 100 mM KCl) and finally resuspended in wash buffer to 20 mg ml⁻¹ lipid concentration. Liposomes were diluted 100-fold in outside buffer

(20 mM HEPES pH 7.4, 100 mM NaCl) and 100 μl aliquots were placed in the wells of a black, flat bottom 96 well plate (Thermo Fischer Scientific) using the software SparkControl v3.2. Fluorescence was measured on Tecan Spark instrument (Tecan Life Sciences) in cycles every 4 seconds. For calcein, the direct quenching by the substrate was measured (excitation: 492 nm and emission: 518 nm), for Fura-2 the ratio between bound (excitation: 340 nm, emission: 510 nm) and unbound state (excitation: 380 nm, emission: 510 nm) was monitored. After 19 cycles, 100 nM valinomycin was added to the liposomes to establish an inside negative membrane potential, after a total of 49 cycles, the substrates were added at indicated concentrations to initiate transport and after a total of 124 cycles, 100 nM of the ionophore calcimycin was added as a control.

To monitor the proton transport, liposomes were diluted 2-fold in ACMA-buffer (10 mM HEPES pH 7.0 or 7.4, 100 mM KCl), subjected to 3 freeze-thaw cycles, then ACMA was added to the liposomes to a final concentration of 50 μM. The liposomes were diluted 100-fold in outside buffer at either pH 7.4, 7.0, or 6.5, and 100 μl aliquots were placed into the wells of a black, flat bottom 96 well plate. Fluorescence was measured on a Tecan Spark every 4 seconds and the direct quenching of ACMA was recorded (excitation: 412 nm, emission: 482 nm). After 19 cycles, the substrate was added to the liposomes, after a total of 49 cycles valinomycin was added to establish an inside negative membrane potential, and after a total of 124 cycles CCCP was added as a control. For all assays, 1 μl of substrates and ionophores was added to each well to avoid dilution effects and mixed 5 times by pipetting.

Data was normalized to cycle 50, and for the FURA-2 data the moving average over 5 data points was calculated. Transport kinetics was determined by fitting the Michaelis-Menten equation to the initial transport rate calculated by linear regression over 16 measurement points (about 1 min), starting 30 seconds after the addition of the substrate after the baseline (no substrate) was subtracted from each measurement (Supplementary Fig. 1b).

### Statistics and reproducibility

Transport data were generally obtained from two independently prepared samples and four technical replicates per reconstitution, yielding a total of eight measurements per data point unless stated otherwise. Values were averaged and displayed errors of individual conditions are s.e.m.. Fits to a Michaelis Menten equation shown in Supplementary Table 1 were obtained from the average of baseline subtracted individual measurements. The error in this case is the standard deviation (s.d.) of the fit. P-values were obtained from fits of individual replicates covering the indicated metal ion concentration range to a Michaelis Menten equation. Averaged $K_m$ values and errors (s.d.) of individual substrate-protein pairs were compared using a two-tailed unpaired T-test with Welch correction (Supplementary Table 2). Statistical analysis was performed using GraphPad Prism 10.

### Reporting summary

Further information on research design is available in the Nature Portfolio Reporting Summary linked to this article.

## Data availability

Coordinates of structures described in this manuscript have been deposited with the PDB (https://www.rcsb.org/) under accession codes 9F6N (DMT1/Sb1), 9F6O (DMT1/Sb1/Sb2), 9F6P (NRAMP1$^{inw}$), 9F6Q (NRAMP1$^{occ}$). Cryo-EM maps have been deposited with the EMDB under accession codes EMD-50235 (DMT1/Sb1), EMD-50236 (DMT1/Sb1/Sb2), EMD-50237 (NRAMP1$^{inw}$), EMD-50238 (NRAMP1$^{occ}$). Previously published structures that were used in this study were deposited under the accession codes 5M87 (EcoDMT) 6TL2 (EcoDMT-Br-BIT complex), 8E6I (DraNRAMP$^{inw}$), 8E6O (DraRNAMP$^{occ}$) and 8E6N (DraNRAMP$^{outw}$). Source data for transport assays, SPR, SEC and SDS-PAGE are provided with this paper. Source data are provided with this paper.

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

## Acknowledgements

Cryo-EM data were collected at the Center for Microscopy and Image Analysis (ZMB) of the University of Zurich. We thank Piotr Szwedziak for support during microscope operation and Markus Seeger for providing access to the sybody libraries. Jens Sobek is acknowledged for help with surface plasmon resonance experiments recorded at the Functional Genomics Center of UZH/ETH Zurich, Elena Lehmann for assistance in data collection and all members of the Dutzler for their help at various stages of the project. This research was supported by a grant of the NCCR TransCure to R.D.

## Author contributions

M.L. cloned, expressed, and purified proteins, performed transport assays, prepared samples cryo-EM, collected and processed cryo-EM data, and built models. C.M. has cloned constructs and carried out initial expression studies. A.F.I. has assisted in protein preparation and transport assays. M.L., C.M. and R.D. jointly planned experiments, analyzed data, and wrote the manuscript.

## Competing interests

The authors declare no competing interests.
