## [Transparent Peer Review file · Nature Communications]

Structural basis for metal ion transport by the human SLC11 proteins DMT1 and NRAMP1

Corresponding Author: Professor Raimund Dutzler

Version 0:

Reviewer comments:

Reviewer #1

(Remarks to the Author)

Liziczai et al report the structures of the human divalent metal-ion transporter-1 (DMT1) and natural resistance-associated macrophage protein-1 (NRAMP1) and their functional characterization by measuring H⁺ or metal-ion transport in reconstituted proteoliposomes with the aid of pH- or metal-sensitive fluorophores.

The structures are novel—the first human structures to be described—and could provide important new information that will guide structure–function analyses and drug design.

The functional data for the most part are not novel, at least for DMT1 for which H⁺/Fe²⁺ symport and uncoupled fluxes has been demonstrated by several other groups. Of potential interest, the authors reach the conclusion that NRAMP1 is also a H⁺/Fe²⁺ symporter whereas one previous report had proposed NRAMP1 to be a H⁺/Fe²⁺ antiporter. Regrettably, the functional experiments and their analyses have not been rigorously performed. The data set suffers from low sample sizes or a reliance on a single observation and any replicate is unreported. Regression analyses are reported without goodness-of-fit estimates or P values. Estimates of apparent affinities and so on are reported without appropriate estimates of uncertainty (e.g. SEM or CI). The authors present conclusions from comparisons (e.g. estimates of relative substrate selectivity between the two proteins) in the absence of any statistical test. The “Statistics and reproducibility” section is a capitulation. Power is not discussed that serves to the statistical analyses are woefully inadequate but still want the reader to believe their assurance that they have used sufficient replicates (which are unreported).

The functional data are overstated and overhyped whereas the novel structural data (much of which is relegated to supplemental data) is not adequately reported or discussed. The potentially interesting conclusions from this investigation relate to the structural evidence underlying metal-ion selectivity between the prokaryote and human DMT1 paralogs in the metal-ion binding site and an explanation for the observed Ca inhibition of metal-ion transport in human DMT1 but the associated functional data are weak.

That the study limitations are not discussed is one thing but that they are overlooked by the authors in reaching overall conclusions is another. For example, only two structures are reported for DMT1 and they are conformationally very similar, so the notion that they reveal the “structural basis for metal ion transport” is an overreach. The investigators have missed the opportunity to examine substrate binding (divorced from the transport cycle) or to provide a more complete model of metal ion transport.

Specific comments for revision

1. The study suffers from a lack of a clearly stated hypothesis. The introduction is overwritten, reads like a short review, and does not articulate the gaps in knowledge that this study was to address. Just one example (ll. 106-109): “Likewise, we tested whether we would find a similar competition of Mn²⁺ and Fe²⁺ transport by the alkaline earth metal ions Ca²⁺ and Mg²⁺, both of which did not show pronounced transport in DMT1 in earlier investigations[4,11].” Ref. 4 did not test calcium transport. Ref. 11 provides no support for Ca²⁺ transport (so “did not show pronounced transport” is odd). So what was the impetus for testing Ca²⁺ and Mg²⁺ transport here? L. 118 might suggest it was to “refute” some previous study but, if there is one, it was not cited. Some prior studies relating to functional studies of SLC11 proteins are not adequately attributed or are imprecisely cited even though they demonstrated transport mechanisms that the present study only confirms. The

organization and language implies that the present study makes novel conclusions about DMT1 or NRAMP1 functional properties when in fact they are largely confirmatory.

2. The conclusion of functional coupling in a symporter or cotransporter requires the demonstration of A-dependent B transport and B-dependent A transport. The authors show metal-ion concentration dependent acidification but not pH-dependent metal-ion transport.

3. Throughout the main body, extended, and supplemental data, the statistical analysis was either minimal or not present. For example, a Michaelis-Menten function was fit to the change in fluorescence as a function of Mn concentration; however, there is no indication on the variability of the regression (residuals of the regression, interpreted from the R^2) or the significance of the fit. Moreover, the K_m for Mn and Fe in DMT1 are compared to the Mn and Fe K_m s for NRAMP1 without any supporting statistical analysis. This could be corrected by inclusion of these comparisons in addition to the interpretation of the magnitude of change for substrate affinity. Similarly, it is unclear what the sample size is for each experiment since it is not indicated in the figure legends and the statistical analysis section suggests multiple replicates with no further information on whether these are biological or technical replicates.

4. How the NRAMP1 structure was generated is poorly explained. The authors describe how the relatively small monomeric proteins make small particle alignment difficult, prompting the use of sybodies in the generation of cryo-EM structure but NRAMP1 did not interact with these. How did the process differ for DMT1 and NRAMP1 structures?

5. Fig 7A depicts an oversimplified transport mechanism that includes a proton leak but not an uncoupled Fe^{2+} as has been previously described in one study and supported by empirical data in other studies even if not discussed there. The schematic proposes three actions, (1) Binding of H^+ within the substrate binding pocket, (2) binding of Fe^{2+} within the substrate binding pocket, and (3) a proton leak. The investigators provide structural evidence only for the second of these three.

6. L. 26: "supraphysiological" would be preferred over "unphysiological."

7. L. 38: L. 39: "Within the blood plasma," not "within the body."

8. L. 55: "in contrast to sodium coupled symporters ..." Na^+ -coupled transporters, among them Na^+ /glucose symporters, Na^+ /amino acid transporters, Na^+ /ascorbate transporters characteristically exhibit Na^+ leaks (uniport) in varying degrees.

9. L. 90: The authors chose to study metal ion transport at neutral pH. That's an odd starting point for known H^+ -coupled transporters, so why? Please justify.

10. LI. 90-92: "In all cases, the time and metal ion concentration-dependent quenching of calcein trapped inside the liposomes underlines their specific uptake," is vague at best, meaningless at worst.

11. The investigators are not first to use these particular fluorophores but they ought to show that, in the preparations used in this study, the fluorophores are reporting metal ion or proton concentrations, e.g. by rapid permeabilization of the proteoliposomes.

12. The authors provide no evidence of an initial linear phase. Quenching looks decidedly like a first-order decay.

13. LI. 94-96: The conclusion "... somewhat lower K_m values ... indicating a higher affinity for $[Fe^{2+}]$ " is not supported by any statistical analysis. L. 96: The statement that Fe^{2+} , "is the primary substrate of both transporters," is odd since only two metals were tested.

14. LI. 96-99: The authors state that the Fe/Mn transport properties of the human SLC11 transporters "closely resemble those of the prokaryotic homologue EcoDMT ..." They do not describe how, they do not describe the properties of EcoDMT, and no analysis is presented.

15. L. 102: The authors have provided no evidence for competition, so the statement "suggesting that both ions could compete for the same binding site," is unsupported. L. 110: "inhibition" not "competition."

16. The nomenclature of assigning concentrations or which ions are being investigated are not intuitive. For example, ratios of Mg to Mn concentrations are used for Figure 1G but in Figure 2A and 2B, a ratio of H^+ to Mn is used. In the case of Figure 2, the assays were performed using the ACMA fluorescent dye. This could be clarified by explicitly stating what the fluorescence is attributed to (transition metal-ion or protons) and subsequently labeling the various conditions of the independent variable.

17. In lines 238 to 241, would it be more accurate to describe NRAMP1_{inw} as a partial inward conformation since the protein has not fully adopted an inward-facing conformation? Along these lines these particular conformations may represent a gradation of the transport mechanism steps. Although transport mechanisms can be simplified to discrete steps there are likely smaller, more granular steps as the substrate moves across the membrane.

18. LI. 392-395 appear contradictory. For example, we get the impression that DMT1 possesses Ca transport function and that certain mutations ablate this function; however, the data do support Ca transport, rather, inhibition of DMT1 via

potentially stabilizing an occluded conformation due to the presence of negatively charged residues at the extracellular face.

19. "Either" is used incorrectly throughout when "both" is needed. (This is one of several English usage error that need to be fixed.)

Reviewer #2

(Remarks to the Author)

The manuscript by Marton Liziczai et al reports structure determinations of human divalent metal ion transporters in the SLC11 family, DMT1 and NRAMP1, and functional characterizations of the two transporters using a liposome flux assay. The structures allow visualization of the overall structures of the transporters, and the identification of a binding site for the divalent transition metal ions. Functional studies show that the purified target proteins mediate co-transport of both proton and transitional divalent cations and that both have a preference for transitional metal ions over calcium or magnesium ions. DMT1 seems sensitive to calcium block, while NRAMP1 is not. In addition, structure of NRAMP1 was determined in two conformations, which show significant movement of TM1a and modest changes in TMs 4 and 5. These new and significant information, in combination with existing structures of bacterial homologs of NRAMP1, provide a more comprehensive and deeper understanding of substrate recognition and transport of human SLC11 family of transporters.

Data acquisition and analysis for structure determinations and functional characterizations are done with care, and the main conclusions are supported by the data. The authors used sybodies to facilitate structure determination of DMT1, and used various controls and an inhibitor TMBIT in the functional assays to enhance the rigor.

I feel that the authors could do more to clarify the proton pathway, although I should clarify that the suggestions listed here should not be taken as a requirement for the current manuscript. The authors indicate that H296 (DMT1 number) could mediate coupled proton transport, and that the salt bridge pairs, E183/R440 and E193/D190/R445, may be involved in either the coupled or non-coupled proton transport or both. Could the the proton transport assay be used on relevant mutations to support these claims? Along the same line of reasoning, could metal ion binding affinity be measured in different pH with the wild type and the H296 mutant?

I also have a request for Figure 5b and f. Could the authors display the density for residues that coordinate the metal ion? That way the sigma levels of the metal ion can be directly compared to protein density in the same map.

Reviewer #3

(Remarks to the Author)

In this work, the authors present the cryo-EM structures of two human SLC11 paralogs, DMT1 and NRAMP1, which are both capable of transporting manganese and iron across the membrane in a manner coupled to the cotransport of protons. Whereas the former structure was determined in complex with two nanobodies selected from synthetic libraries, the latter was solved in isolation. They also developed the biochemical assay system for assessing the Fe²⁺/Mn²⁺/H⁺ transport properties of the SLC11 paralogs. The present cryo-EM analysis revealed the first near-atomic resolution structures of DMT1 and NRAMP1 in two different conformations, which significantly advanced our understanding of conformational transitions during their heavy metal ion transport. Nevertheless, important questions remain to be uncovered especially about a mechanism of coupling between the metal and proton transports exerted by these transporters. Overall, structural and biochemical analyses have been nicely done. However, some conclusions do not seem to be well supported by experimental data and figures. Some parts of the text are not clearly written. In these regards, this manuscript is still premature. Without addressing all the following issues properly, this reviewer could not recommend this paper for publication in this journal.

Major issues:

- 1) The authors believe that both DMT1 and NRAMP1 are H⁺-coupled symporters. According to the text, they assayed the Mn²⁺ and Fe²⁺-dependent acidification of vesicles for confirmation. However, only Mn²⁺-coupled H⁺ transport is shown in Fig. 2. Data needs to be shown also with Fe²⁺.
- 2) Fig. 4b is very unclear and needs to be amended so that we can easily grasp the position and orientation of each TM helix in the three-dimensional structure of DMT1. For this purpose, the label should be added for each TM helix, and another angle view (maybe top view) should be provided This is the same case with Fig. 4d and 4i. Without additional clear images, it would hard to accept the author's statement that the observed structures of DMT1 and NRAMP1 represent an intermediate on the transport cycle placed in-between the two conformations of the prokaryotic homolog (lines 188-190). Additionally, the a6-a10 segment should be shown in different colors so as to easily distinguish those of DMT1 and DraNRAMP.
- 3) Line 238; Extended Data Fig. 4i is missing. Without the cross section image, it would be hard to see the aqueous access path to the metal ion binding site located in the center of the protein.
- 4) 2nd paragraph on page 14; the description of the presumed H⁺ pathway in DMT1 is very unclear. To make their statement regarding the H⁺ pathway more convincing, the pathway needs to be highlighted with high accuracy. In particular, the locations of His296 and three conserved acidic residues, which are presumed to participate in H⁺ transport, need to be shown clearly not only in whole structure and but also along the H⁺ pathway. It is also important to discuss the position of the presumed H⁺ pathway relative to the metal transport pathway. In this context, the sentences (lines 292-296) are hard to

understand. The substantial text revision is necessary hopefully with a supplementary figure.

5) Biochemical assays using ion binding-site mutants shown in Fig. 6 need to be conducted with Fe²⁺ as well as Mn²⁺. This additional data would enable to discuss whether or not the same amino acids are involved in Fe²⁺ binding and transport as Mn²⁺ by DMT1.

6) Cartoons shown in Fig. 7a are too much simplified. The presented cryo-EM structures provide insight into the TM helix rearrangement during the transition from the occluded to inward/outward-facing states. This information should be included in this figure panel.

Minor issues:

1) Lines 148-149: The authors state that Sb1 and Sb2 bind to non-overlapping epitopes of DMT1. However, no experimental data are shown to support this conclusion. Some supportive data should be added.

2) It is very exciting that the authors succeeded in cryo-EM structure analysis of ~52-kDa NRAMP1 without binder proteins although DMT1 with similar size and overall structure to NRAMP1 requires a nanobody for near-atomic resolution cryo-EM analysis. It would be reader-friendly and beneficial for cryo-EM scientists to additionally discuss the reason why the cryo-EM analysis was successful for NRAMP1 in isolation, which is not the case with DMT1.

3) Fig.3c is very unclear. Although the authors point out conformational changes in the extracellular part of alpha6a, the figure panel does not clearly show the location of alpha6a-helix in the whole structure. Figure revision is needed.

4) Line 14; "has" should read "have".

5) Line 76; "our" should read "this".

6) Line 108; "in DMT1" should read "by DMT1".

7) Line 153; insert "resolutions" after "3.6 Å".

8) Lines 156-157; The phrase "conformational heterogeneity" is very ambiguous. Does the Sb2 itself adopts different conformations or bind to different sites of DMT1?

9) Line 239; insert "site" after "binding".

Reviewer #4

(Remarks to the Author)

Version 1:

Reviewer comments:

Reviewer #1

(Remarks to the Author)

The authors have improved the description of statistical methods and reporting, and now provide measures of uncertainty for the parameters estimated (e.g. Km, Ki). I appreciate their intent to interpret functional data (for both proteins and mutants) conservatively given the limitations and challenges of their assays. In their rebuttal, the authors state that "The hypothesis of our study is obvious," and go on to describe the central aim of the study but do not state a hypothesis. To reiterate, the manuscript would benefit from the inclusion of a clearly stated hypothesis in the introduction.

Reviewer #2

(Remarks to the Author)

I have no further comments.

Reviewer #3

(Remarks to the Author)

I believe that the authors have addressed all of my concerns appropriately in the revised version. Only minor revisions seem necessary for acceptance of this paper, as described below.

1) Presumed H⁺ transport pathways in DMT1 are illustrated in newly prepared Fig. 5H. Although this added figure is highly beneficial for readers, the left panel displaying the overall structure is still hard to grasp due to too small size, too small and

imprecisely positioned labels, and the shaded square. It would better that the shaded square is replaced by a blank square, and that residues involved in the H⁺ transport are highlighted in the right inset with an outer frame line.

2) In supplementary Fig. 11c-l, the labels "Mn²⁺" on the right should be amended to "Fe²⁺".

We thank the reviewers for their generally constructive and insightful comments, which we have considered in our revised manuscript and addressed in detail below. Before responding to reviewer comments, we would like to emphasize important aspects related to this work, which might not always have received sufficient attention.

This manuscript summarizes over four years of intense research on a topic my group has worked on during the last 15 years, the structure and function of the SLC11 family of transition metal ion transporters. The first breakthrough in our studies was reached about 10 years ago by the structural and functional characterization of the prokaryotic homologue ScaDMT, which has defined the general architecture of the family in an inward-facing conformation, and the underlying basis of metal ion interactions¹. Three years later, this first study was followed up by investigations on a different prokaryotic transporter, EcoDMT, which revealed the structure of an outward-facing conformation and provided insight into the coupling to protons². Following, we have studied the mechanism of inhibition of human and bacterial orthologs³ and the transport properties of family members with altered substrate selectivity^{4,5}. Together with results obtained by others, these studies have provided a solid foundation for our current work. For our previous investigations, we have established an *in vitro* transport assay based on liposome-reconstituted protein and the detection of transported ions by fluorophores, which was used throughout to characterize functional properties. Our studies have now cumulated in the structural and functional characterization of the two human transporters SLC11A1 (NRAMP1) and SLC11A2 (DMT1), which together constitute a highlight of our efforts on the SLC11/NRAMP family. I would like to emphasize the enormous challenges associated with the work on human membrane proteins, which made every step in the process considerably more difficult than equivalent experiments carried out with their prokaryotic homologues. The low yield of only about 100 µg of protein per liter of mammalian cell culture in case of DMT1 and 40 µg in case of NRAMP1 has required the frequent preparation of large suspension cultures and the poor stability of the protein in detergent solution has severely complicated experiments starting from sybody selection, structural characterization and reconstitution. With respect to the structural aspects of the work, we have benefitted from a constant improvement of procedures over the course of the study, which has ultimately allowed us to determine the structure of NRAMP1 in absence of fiducial marker. This was still considered to be out of reach at the time when we have determined the structure of DMT1. We would like to emphasize that all structures presented in our study are still at the edge of what is currently

achievable by cryo-EM, as also acknowledged by reviewer 3. The presented liposome-based functional experiments allow for a direct comparison with previous work on other family members by us and others and they also permit the relation of functional properties between the well-studied DMT1 and the functionally poorly characterized NRAMP1. Here, I want to particularly emphasize the technical challenges associated with the functional investigation of these human transport proteins by *in vitro* studies, which limits the range of investigated conditions. For each reconstitution, we use protein purified from about three liter of HEK cell culture with yields decreasing for certain point mutations (the protein from one liter of expression culture sufficed for the measurement of eight data points with four technical replicates). Nevertheless, we have in essentially each case obtained data from two independent protein preparations with very similar results, underlining the robustness of our data. We also want to point out that our study aims at a semiquantitative characterization of transport where we never aim to make arguments based on small differences in transport properties. Nevertheless, in response to a reviewer request, we have now provided a more thorough statistical analysis in our revision (which did not alter any of our conclusions). The applied assay conditions for transport at neutral pH reflect the optimal sensitivity of the applied fluorophores. These have also been used in previous work and provided robust data on coupled transport. Compared to the assay of transition metal ion transport, the assay of H⁺ transport is considerably more challenging and less sensitive due to the non-specific leak of H⁺ into liposomes, which we have minimized by the careful selection of lipids, and the uncoupled H⁺ transport observed in both human transporters, which is a real feature that has been thoroughly characterized in our study. While we have obtained robust data that demonstrates H⁺ cotransport in both WT proteins, we decided against using the assay for the characterization of mutants for gaining deeper insight into the H⁺ coupling mechanism. This decision was based on initial experiments, which we considered as not fully conclusive as a consequence of the low sensitivity of the assay, which prevents a reliable evaluation of the data.

Our functional data have confirmed several previous observations obtained from the use of orthogonal approaches, which we consider as positive and we made sure to refer to these data accordingly. We also want to emphasize that we always clearly stated where our work is based on such previous studies. Nevertheless, we hope the reviewers will recognize the enormous efforts underlying the current study and the general conclusiveness of our data, which were interpreted

conservatively and have for the first time provided structural and functional data on purified members of this important protein family.

Our revised manuscript contains the following major changes:

- Data on the Fe^{2+} transport properties of mutants showing a similar strong decay of transport as the same experiments carried out with Mn^{2+} displayed in Supplementary Fig 11c-l.
- A thorough statistical analysis of kinetic fits of transport data detailed in Supplementary Tables 1 and 2.
- Experiments investigating the H^+ coupling of Fe^{2+} transport displayed in Supplementary Fig. 2c.
- Experiments demonstrating the simultaneous binding of both sybodies displayed as Supplementary Fig. 3h.
- Modification in several figures including a better documentation of presumed H^+ efflux pathways displayed in Fig. 5h and an additional panel detailing conformational differences between known structures of SLC11 family members displayed in Fig. 7c.
- Changes throughout the text including a condensed introduction and clarifications with respect to the interpretation of results.

Following, we provide a detailed response to reviewer comments.

REVIEWER COMMENTS

Reviewer #1 (Remarks to the Author):

Liziczai et al report the structures of the human divalent metal-ion transporter-1 (DMT1) and natural resistance-associated macrophage protein-1 (NRAMP1) and their functional characterization by measuring H^+ or metal-ion transport in reconstituted proteoliposomes with the aid of pH- or metal-sensitive fluorophores.

The structures are novel—the first human structures to be described—and could provide important new information that will guide structure–function analyses and drug design.

The functional data for the most part are not novel, at least for DMT1 for which H⁺/Fe²⁺ symport and uncoupled fluxes has been demonstrated by several other groups. Of potential interest, the authors reach the conclusion that NRAMP1 is also a H⁺/Fe²⁺ symporter whereas one previous report had proposed NRAMP1 to be a H⁺/Fe²⁺ antiporter. Regrettably, the functional experiments and their analyses have not been rigorously performed. The data set suffers from low sample sizes or a reliance on a single observation and any replicate is unreported. Regression analyses are reported without goodness-of-fit estimates or P values. Estimates of apparent affinities and so on are reported without appropriate estimates of uncertainty (e.g. SEM or CI). The authors present conclusions from comparisons (e.g. estimates of relative substrate selectivity between the two proteins) in the absence of any statistical test. The “Statistics and reproducibility” section is a capitulation. Power is not discussed that serves to the statistical analyses are woefully inadequate but still want the reader to believe their assurance that they have used sufficient replicates (which are unreported).

We have now improved the statistical analysis of the functional data in our revised manuscript, which did in no way alter the conclusions made in the original submission. In general, we disagree with the reviewer challenging the novelty of our functional data. These provide first insight into transport properties of both human family members in a reconstituted system and as such allow the direct comparison with data obtained from prokaryotic homologues, which were assayed in a similar manner. In case of NRAMP1, our data provides the first detailed characterization of transport properties, which were previously inaccessible due to the intracellular localization of the protein (although there is one study using cellular experiments with overexpressed protein, part of which was targeted to the plasma membrane, which we referred to in our manuscript⁶). In case of DMT1, our data generally agrees with previous studies by electrophysiology obtained in the *X. laevis* system, which underlies the validity of the used assay. In combination, our data allows a direct comparison between the two paralogues.

With respect to sample size, we disagree with the claim of the reviewer that the sample size would be too low to reach meaningful conclusions. As stated before, nearly all experiments (particularly all experiments used for a quantitative evaluation) were carried out with samples obtained from two independent reconstitutions (i.e. the equivalent of two biological replicates). With four

technical replicates from each reconstitution, all datapoints show averages from eight measurements. The comparatively small errors, which are displayed in our figures, illustrate the high data quality. The sample sizes were defined in the respective figure legends of our manuscript but might have been overlooked by the reviewer.

We have now provided errors derived from fits of individual measurements to a Michaelis Menten equation and found differences in the obtained values between the three transporters (two human and one bacterial transporter) and between Fe^{2+} and Mn^{2+} transport in most cases as statistically significant. The data is now provided as Supplementary Tables 1 and 2. We also want to emphasize that in our study, we use the kinetic data in a semi-quantitative manner to demonstrate the saturation of transport in different family members at similar micromolar ion concentrations. We have not used this data to make mechanistic claims based on small quantitative differences. In case of binding site mutants, we show a strongly compromised transport behavior that differs markedly from WT. In this case we refrained from any quantitative interpretation.

The functional data are overstated and overhyped whereas the novel structural data (much of which is relegated to supplemental data) is not adequately reported or discussed. The potentially interesting conclusions from this investigation relate to the structural evidence underlying metal-ion selectivity between the prokaryote and human DMT1 paralogs in the metal-ion binding site and an explanation for the observed Ca inhibition of metal-ion transport in human DMT1 but the associated functional data are weak.

We disagree on this point. The bulk of the manuscript describes structural features and structure-function relationships that are based on the novel structures. The functional experiments relate the behavior of both proteins to prokaryotic homologues that were characterized by equivalent methods. The effect of Ca^{2+} in case of DMT1 is robust and well documented. The absent effect of Mg^{2+} in both case is relevant in light of a previous study postulating Mg^{2+} transport of this homologue⁷.

That the study limitations are not discussed is one thing but that they are overlooked by the authors in reaching overall conclusions is another. For example, only two structures are reported for DMT1 and they are conformationally very similar, so the notion that they reveal the “structural basis for metal ion transport” is an overreach. The investigators have missed the opportunity to examine substrate binding (divorced from the transport cycle) or to provide a more complete model of metal ion transport.

It is somewhat difficult to follow the argument of the reviewer, who might not be a structural biologist and thus not appreciate what is within reach in such study. As emphasized before, our manuscript summarizes the results of an extended and biochemically challenging investigation. It describes the first structures of human members of the SLC11 family and provides complementary functional data. It also builds on several previous studies by our lab and others. In combination with previous work the data provide important mechanistic insight into metal ion transport.

Specific comments for revision

1. The study suffers from a lack of a clearly stated hypothesis. The introduction is overwritten, reads like a short review, and does not articulate the gaps in knowledge that this study was to address. Just one example (ll. 106-109): “Likewise, we tested whether we would find a similar competition of Mn²⁺ and Fe²⁺ transport by the alkaline earth metal ions Ca²⁺ and Mg²⁺, both of which did not show pronounced transport in DMT1 in earlier investigations[4,11].” Ref. 4 did not test calcium transport. Ref. 11 provides no support for Ca²⁺ transport (so “did not show pronounced transport” is odd). So what was the impetus for testing Ca²⁺ and Mg²⁺ transport here? L. 118 might suggest it was to “refute” some previous study but, if there is one, it was not cited. Some prior studies relating to functional studies of SLC11 proteins are not adequately attributed or are imprecisely cited even though they demonstrated transport mechanisms that the present study only confirms. The organization and language implies that the present study makes novel conclusions about DMT1 or NRAMP1 functional properties when in fact they are largely confirmatory.

We disagree on several points. The hypothesis of our study is obvious. My group has been interested in the SLC11 family for over 15 years and the aim of this study was to characterize the structure and function of the two human family members and compare these with previous data of prokaryotic homologues. The introduction summarizes these previous findings but we have condensed it to focus on the subject of this study. The mentioned sentence was part of the results and not the introduction and reference 4 has investigated Ca^{2+} transport in DMT1 by electrophysiology but did not identify any transport-related currents⁸ (See Figure 5a of the mentioned study). The investigation of Ca^{2+} transport was carried out to permit a comparison with previous studies of prokaryotic transporters where certain mutants in the ion binding site conferred Ca^{2+} transport^{4,9}. The investigation of Mg^{2+} transport was motivated by a recent study on NRAMP1⁷. We have always referred to previous data on DMT1 where appropriate and there was no direct transport data of NRAMP1 available except for one study in CHO cells, where a fraction of the overexpressed protein was targeted to the plasma membrane and that was referred to in our manuscript⁶.

2. The conclusion of functional coupling in a symporter or cotransporter requires the demonstration of A-dependent B transport and B-dependent A transport. The authors show metal-ion concentration dependent acidification but not pH-dependent metal-ion transport.

We are fully aware of the functional implication of coupling. In case of our study, we are limited by the tightness of liposomes for H^+ , the sensitivity of used fluorophores and uncoupled H^+ leaks, which prevent the accumulation of metal ion transport by a proton gradient. Such limitations are generally accepted and have been discussed in previous studies². H^+ coupling of Fe^{2+} transport has also been demonstrated previously and we refer to these studies in our manuscript^{8,10}.

3. Throughout the main body, extended, and supplemental data, the statistical analysis was either minimal or not present. For example, a Michaelis-Menten function was fit to the change in fluorescence as a function of Mn concentration; however, there is no indication on the variability of the regression (residuals of the regression, interpreted from the R^2) or the significance of the fit. Moreover, the K_m for Mn and Fe in DMT1 are compared to the Mn and

Fe Kms for NRAMP1 without any supporting statistical analysis. This could be corrected by inclusion of these comparisons in addition to the interpretation of the magnitude of change for substrate affinity. Similarly, it is unclear what the sample size is for each experiment is since it is not indicated in the figure legends and the statistical analysis section suggests multiple replicates with no further information on whether these are biological or technical replicates.

We now provide a statistical analysis of the fits and analyzed differences between distinct conditions (Supplementary Tables 1 and 2). We and others have used this assay previously in numerous publications, where we have described the linear relationship of the metal ion concentration to the quenching of the fluorophore in the investigated range^{2-5,9,11-13}. A panel illustrating this relationship in the measured range is provided in Supplementary Fig. 1b. The number of repeats is clearly stated in the legends, which also define that essentially all relevant data points are averages from eight replicates from two independent reconstitutions (corresponding to four technical replicates for each reconstitution). We also want to point out again that we have interpreted the results conservatively. Although the lower K_m values for both Fe^{2+} and Mn^{2+} in NRAMP1 compared to DMT1 are statistically significant, we refrained from a detailed discussion of these properties and restricted the conclusions to the fact that both proteins show similar substrate preference. In case of point mutations in the ion binding site, the associated effects on transport were severe and we restricted ourselves to a qualitative interpretation in light of limitation of the applied assay.

4. How the NRAMP1 structure was generated is poorly explained. The authors describe how the relatively small monomeric proteins make small particle alignment difficult, prompting the use of sybodies in the generation of cryo-EM structure but NRAMP1 did not interact with these. How did the process differ for DMT1 and NRAMP1 structures?

Data processing and reconstruction is outlined in detail in Supplementary Fig. 6 and in the method section, which provides more details than similar manuscripts and contains sufficient information to be able to follow the structure determination process.

We have also included the following sentence in the results, line 167-171:

As in case of NRAMP1, the low yield and poor stability of the protein in detergent solution has complicates sybody selection, we have attempted structure determination without fiducial markers by combining 57'783 micrographs from five datasets. These data have allowed us to identify two distinct conformations of the transporter in the same sample (Fig. 3d-g, Supplementary Fig. 6, Supplementary Table 3).

5. Fig 7A depicts an oversimplified transport mechanism that includes a proton leak but not an uncoupled Fe²⁺ as has been previously described in one study and supported by empirical data in other studies even if not discussed there. The schematic proposes three actions, (1) Binding of H⁺ within the substrate binding pocket, (2) binding of Fe²⁺ within the substrate binding pocket, and (3) a proton leak. The investigators provide structural evidence only for the second of these three.

We did not intend to provide a detailed kinetic mechanism of transport in Fig. 7a. Instead, we wanted to sketch an overview of the findings obtained in our study for the discussion. We have thus reworded the title transport mechanism to transport properties to make this clearer.

6. L. 26: “supraphysiological” would be preferred over “unphysiological.”

This part was removed from the introduction.

7. L. 38: L. 39: “Within the blood plasma,” not “within the body.”

We have condensed and reworded this part of the introduction.

Line 34-35:

Within the body, DMT1 resides within endosomes to mediate the release of endocytosed Fe²⁺ into the cytoplasm powered by the proton gradient that is established by V-type ATPases.

8. L. 55: “in contrast to sodium coupled symporters ...” Na^+ -coupled transporters, among them Na^+ /glucose symporters, Na^+ /amino acid transporters, Na^+ /ascorbate transporters characteristically exhibit Na^+ leaks (uniport) in varying degrees.

Since we do not want to engage in a detailed discussion of uncoupled Na^+ leaks, we have modified our wording.

Line 49-51:

As a secondary active transporter, DMT1 utilizes a proton gradient to concentrate iron inside cells beyond its electrochemical equilibrium. However, the coupling is not strict, as protons were also described to pass the protein without being coupled to metal ion transport and *vice versa* uncoupled metal ion transport was observed at neutral pH.

9. L. 90: The authors chose to study metal ion transport at neutral pH. That’s an odd starting point for known H^+ -coupled transporters, so why? Please justify.

The choice of the pH was dictated by the sensitivity of the used fluorophore calcein and the increased leakage of H^+ at lower pH values. We and others have used equivalent conditions in previous studies^{2,4,5,9,13}. Despite the chosen pH, we observe robust metal ion transport, which in case of Mn^{2+} also leads to a concentration-dependent acidification of vesicles underlining substrate coupling even in neutral conditions.

We have introduced the following sentence (Line 82-84)

After their reconstitution into proteoliposomes, we have assayed the ability of either protein to mediate the transport of the transition metal ions Mn^{2+} and Fe^{2+} , using the fluorophore calcein. A neutral pH was chosen to optimize calcein sensitivity.

10. Ll. 90-92: “In all cases, the time and metal ion concentration-dependent quenching of calcein trapped inside the liposomes underlines their specific uptake,” is vague at best, meaningless at worst.

We have revised the sentence to (Line 84-86):

In all cases, we observe a metal ion concentration-dependent quenching of the fluorophore trapped inside the liposomes as a consequence of their uptake.

11. The investigators are not first to use these particular fluorophores but they ought to show that, in the preparations used in this study, the fluorophores are reporting metal ion or proton concentrations, e.g. by rapid permeabilization of the proteoliposomes.

This has been done for every transport experiment but is not shown in the main figures to provide a better view of transport. We have added full traces of the experiment including the equilibration of metal ions after addition of calicmycin in the supplementary data (Supplementary Figs. 1c-f, 2a, b, and 11a, b, e, f, g, i, j, l).

12. The authors provide no evidence of an initial linear phase. Quenching looks decidedly like a first-order decay.

Examples for the essential linearity of the process are now demonstrated in Supplementary Fig. 1b as blowup of the region of transport that was used for data fitting. This panel displays a quasi-linear region reached 30 sec after addition of the metal ion. The procedure was described in the methods and the data does not alter significantly upon change of the interval used for evaluating slopes.

13. Ll. 94-96: The conclusion "... somewhat lower K_m values ... indicating a higher affinity for $[Fe^{2+}]$ " is not supported by any statistical analysis. L. 96: The statement that Fe^{2+} , "is the primary substrate of both transporters," is odd since only two metals were tested.

We now provide a statistical analysis in Supplementary Tables 1 and 2 showing that the difference is significant in case of DMT1 but not NRAMP1.

We have also altered the sentence to:

Line 89-92

In case of Fe^{2+} , we find similar transport properties with somewhat lower K_m values of 2.5 μM for DMT1 and 1.4 μM for NRAMP1, indicating a higher affinity for this transition metal ion, which is the primary substrate of both transporters in a physiological setting, although in case of NRAMP1, the differences in the K_m between both ions measured with this *in vitro* assay are not statistically significant (Fig. 1c, d, Supplementary Tables 1 and 2).

14. Ll. 96-99: The authors state that the Fe/Mn transport properties of the human SLC11 transporters “closely resemble those of the prokaryotic homologue EcoDMT ...” They do not describe how, they do not describe the properties of EcoDMT, and no analysis is presented.

Data for EcoDMT showing a similar Mn^{2+} and Fe^{2+} transport properties are shown in Supplementary Fig. 1g, h and have been investigated in detail in previous studies^{2,4}. A statistical analysis is provided in Supplementary Tables 1, 2.

15. L. 102: The authors have provided no evidence for competition, so the statement “suggesting that both ions could compete for the same binding site,” is unsupported. L. 110: “inhibition” not “competition.”

While strictly speaking the reviewer might be correct, there is ample of evidence for a competition between Mn^{2+} and Cd^{2+} for the same binding site. Transport of Cd^{2+} is demonstrated in Supplementary Fig. 1i and was previously shown for DMT1 (ref. ^{1,8,14}) and its prokaryotic homologues^{1,9,13}. Additionally, its binding to the same site that is also occupied by Mn^{2+} and Fe^{2+} in prokaryotic homologues^{1,13}, which justifies the assumption that the inhibition is a consequence of competition.

We have reworded the sentence to:

Line 97-102

We have also investigated whether the addition of Cd^{2+} , which does not quench calcein fluorescence, would interfere with Mn^{2+} uptake and found a nearly complete suppression of transport at low μM concentrations in both human paralogs, suggesting that the two metal ions could compete for the same binding site (Fig. 1e). Such competition is supported by structures of prokaryotic homologues of the SLC11 family where Cd^{2+} was shown to occupy the same metal ion binding site as Mn^{2+} and Fe^{2+} .

16. The nomenclature of assigning concentrations or which ions are being investigated are not intuitive. For example, ratios of Mg to Mn concentrations are used for Figure 1G but in Figure 2A and 2B, a ratio of H^+ to Mn is used. In the case of Figure 2, the assays were performed using the ACMA fluorescent dye. This could be clarified by explicitly stating what the fluorescence is attributed to (transition metal-ion or protons) and subsequently labeling the various conditions of the independent variable.

We have changed the subtitles in Fig. 2 and Supplementary Fig. 2 to H^+ - Mn^{2+} or H^+ - Fe^{2+} cotransport.

17. In lines 238 to 241, would it be more accurate to describe $\text{NRAMP1}^{\text{inw}}$ as a partial inward conformation since the protein has not fully adopted an inward-facing conformation? Along these lines these particular conformations may represent a gradation of the transport mechanism steps. Although transport mechanisms can be simplified to discrete steps there are likely smaller, more granular steps as the substrate moves across the membrane.

While for simplicity we prefer to keep the term $\text{NRAMP1}^{\text{inw}}$, we have made it clear in several places that this conformation has not fully opened the ion access pathway to the inside (see also the revised Fig. 2j and the new panel Fig. 7c, which defines the relationship between known conformations). While the transition between inward and outward-facing states is continuous, there would still be a limited number of local energy minima, which define structural intermediates on the pathway.

18. Ll. 392-395 appear contradictory. For example, we get the impression that DMT1 possesses Ca transport function and that certain mutations ablate this function; however, the data do support Ca transport, rather, inhibition of DMT1 via potentially stabilizing an occluded conformation due to the presence of negatively charged residues at the extracellular face.

We have never claimed Ca²⁺ transport for WT proteins and made clear that the effect in case of DMT1 is due to inhibition, which could be mediated by three negatively charged residues depicted in Figs. 4f and 7b. In the specified lines we are referring to a mutation of the conserved binding site methionine to alanine, which in case of the prokaryotic transporters EcoDMT and DraNRAMP has yielded a protein that was capable to transport Ca²⁺ (ref. 4,9). No such transport was observed for the equivalent mutation in DMT1 (Supplementary Fig. 11m, n).

19. “Either” is used incorrectly throughout when “both” is needed. (This is one of several English usage error that need to be fixed.)

We have introduced the correction. Eventual remaining errors will be corrected in the editorial process.

Reviewer #2 (Remarks to the Author):

The manuscript by Marton Liziczai et al reports structure determinations of human divalent metal ion transporters in the SLC11 family, DMT1 and NRAMP1, and functional characterizations of the two transporters using a liposome flux assay. The structures allow visualization of the overall structures of the transporters, and the identification of a binding site for the divalent transition metal ions. Functional studies show that the purified target proteins mediate co-transport of both proton and transitional divalent cations and that both have a preference for transitional metal ions over calcium or magnesium ions. DMT1 seems sensitive to calcium block, while NRAMP1 is not. In addition, structure of NRAMP1 was determined in two conformations, which show significant movement of TM1a and modest changes in TMs 4

and 5. These new and significant information, in combination with existing structures of bacterial homologs of NRAMP1, provide a more comprehensive and deeper understanding of substrate recognition and transport of human SLC11 family of transporters.

Data acquisition and analysis for structure determinations and functional characterizations are done with care, and the main conclusions are supported by the data. The authors used sybodies to facilitate structure determination of DMT1, and used various controls and an inhibitor TMBIT in the functional assays to enhance the rigor.

I feel that the authors could do more to clarify the proton pathway, although I should clarify that the suggestions listed here should not be taken as a requirement for the current manuscript. The authors indicate that H296 (DMT1 number) could mediate coupled proton transport, and that the salt bridge pairs, E183/R440 and E193/D190/R445, may be involved in either the coupled or non-coupled proton transport or both. Could the the proton transport assay be used on relevant mutations to support these claims? Along the same line of reasoning, could metal ion binding affinity be measured in different pH with the wild type and the H296 mutant?

The investigation of H⁺ transport in the SLC11 family is a challenging task which, when using the applied methods, is limited by the sensitivity of the assay and the presence of uncoupled H⁺ leaks. We and others have previously studied H⁺ transport in prokaryotic homologues, which share generally similar mechanisms and structural features^{2,4,11,12,15}. The reference to the specified residues in both human transporters is based on these studies. We tried to investigate the effect of equivalent mutants in DMT1, but while we found some effects pointing in the expected direction, we do not consider the results sufficiently conclusive to justify their inclusion in the paper. We have also carried out transport experiments at different pH but found any specific effect be covered by the stronger proton leak and lower fluorophore sensitivity at lower pH and thus also refrained from including such experiments. The effect of the mutant H296A on Mn²⁺ and H⁺ transport is shown in Fig. R1 where we observe robust Mn²⁺ transport and where the presence of coupled H⁺ transport unclear. As the detailed mechanism of H⁺ coupling is even in prokaryotic transporters not fully understood, we decided not to address this important property in our manuscript in greater detail.

Figure R1 | Transport properties of the DMT1 mutant H296A. **a** Ribbon representation of the unwound region of α -helix 6 with side chains of the binding site residue Met 249 and the conserved His 296 shown as sticks. **b** Mn^{2+} transport into proteoliposomes containing the DMT1 mutant H296A, assayed by the quenching of the fluorophore calcein trapped inside vesicles. Time and concentration dependence of metal ion uptake is shown on the left. Right panels show fits of the Michaelis-Menten equation to initial transport rates with respective K_m values indicated. **c** Metal dependent H^+ transport into the same proteoliposomes assayed with the fluorophore ACMA show an unclear transport phenotype. **b, c** Data show mean of eight experiments from two independent reconstitutions, errors are s.e.m.. Trace of WT upon addition of 100 and 50 μM Mn^{2+} for calcein and ACMA assays, respectively, are shown as dashed line for comparison.

I also have a request for Figure 5b and f. Could the authors display the density for residues that coordinate the metal ion? That way the sigma levels of the metal ion can be directly compared to protein density in the same map.

Such representation was previously provided in Supplementary Fig. 9 and has now been included as panels c and h to Figure 5.

Reviewer #3 (Remarks to the Author):

In this work, the authors present the cryo-EM structures of two human SLC11 paralogs, DMT1 and NRAMPI, which are both capable of transporting manganese and iron across the

membrane in a manner coupled to the cotransport of protons. Whereas the former structure was determined in complex with two nanobodies selected from synthetic libraries, the latter was solved in isolation. They also developed the biochemical assay system for assessing the Fe²⁺/Mn²⁺/H⁺ transport properties of the SLC11 paralogs. The present cryo-EM analysis revealed the first near-atomic resolution structures of DMT1 and NRAMP1 in two different conformations, which significantly advanced our understanding of conformational transitions during their heavy metal ion transport. Nevertheless, important questions remain to be uncovered especially about a mechanism of coupling between the metal and proton transports exerted by these transporters. Overall, structural and biochemical analyses have been nicely done. However, some conclusions do not seem to be well supported by experimental data and figures. Some parts of the text are not clearly written. In these regards, this manuscript is still premature. Without addressing all the following issues properly, this reviewer could not recommend this paper for publication in this journal.

We hope to have addressed the comments by the reviewer in a satisfying manner.

Major issues:

1) The authors believe that both DMT1 and NRAMP1 are H⁺-coupled symporters. According to the text, they assayed the Mn²⁺ and Fe²⁺-dependent acidification of vesicles for confirmation. However, only Mn²⁺-coupled H⁺ transport is shown in Fig. 2. Data needs to be shown also with Fe²⁺.

While we have assayed Fe²⁺-dependent H⁺ transport in a similar manner as for Mn²⁺, the results were in this case less clear, which we attribute to the poor sensitivity of the assay but which could also reflect a lack of coupling to protons at neutral pH that was reported in a previous study for Fe²⁺ (ref. ¹⁰). Experiments are shown in Fig R2 (with panels a and b also displayed as Supplementary Fig 2c). A small increase in acidification in presence of Fe²⁺, which is reflected in an enhanced fluorescence decline is observed at symmetric pH7 (Fig. R2a), which is further increased upon lowering the outside pH to 6.5 (Fig. R1b). In case of a pH decrease of the outside medium to 5.5, the uncoupled leaks become dominant and cover any metal driven H⁺ symport

(Fig. R2c). We have changed the text to specify that H⁺ coupling was shown conclusively only for Mn²⁺.

Line 122-126

In these experiments, we find a robust metal ion concentration-dependent quenching of the fluorophore, emphasizing that Mn²⁺ transport is coupled to H⁺ even at neutral pH (Fig. 2a-c, Supplementary Fig. 2a, b). In case of Fe²⁺, the concentration-dependent acidification is less obvious, which could either reflect the limited sensitivity of the assay or a previously reported uncoupled transport of this metal ion at elevated pH.

Figure R2 | H⁺ coupled Fe²⁺ transport in DMT1 investigated by an *in vitro* assay. H⁺ transport is assayed with the fluorophore ACMA. **a-c** H⁺ transport into proteoliposomes containing DMT1 upon addition of Mn²⁺ at an outside pH of 7.0 (**a**) 6,5 (**b**) or 5.5 (**c**). The initial pH inside proteoliposomes was set to 7.0. Data show mean of four experiments from one reconstitution, errors are s.e.m. Fluorescence is normalized to the value after addition of substrate (t=0). Applied metal ion concentrations are indicated.

2) Fig. 4b is very unclear and needs to be amended so that we can easily grasp the position and orientation of each TM helix in the three-dimensional structure of DMT1. For this purpose, the label should be added for each TM helix, and another angle view (maybe top view) should be provided This is the same case with Fig. 4d and 4i. Without additional clear images, it would hard to accept the author's statement that the observed structures of DMT1 and NRAMP1

represent an intermediate on the transport cycle placed in-between the two conformations of the prokaryotic homolog (lines 188-190). Additionally, the a6-a10 segment should be shown in different colors so as to easily distinguish those of DMT1 and DraNRAMP.

We have revised the figure panels b, e and i to show views from within the membrane and from the cytoplasm and included helix labels. Please note that the correspondence between conformations is shown in detail on the basis of single helices in Supplementary Fig. 9. We have also changed the color scheme of DraNRAMP to facilitate comparison with the human transporters.

3) Line 238; Extended Data Fig. 4i is missing. Without the cross section image, it would be hard to see the aqueous access path to the metal ion binding site located in the center of the protein.

This was a mistake in the figure reference. The correct reference is Fig. 4c, which was corrected in the manuscript.

4) 2nd paragraph on page 14; the description of the presumed H⁺ pathway in DMT1 is very unclear. To make their statement regarding the H⁺ pathway more convincing, the pathway needs to be highlighted with high accuracy. In particular, the locations of His296 and three conserved acidic residues, which are presumed to participate in H⁺ transport, need to be shown clearly not only in whole structure and but also along the H⁺ pathway. It is also important to discuss the position of the presumed H⁺ pathway relative to the metal transport pathway. In this context, the sentences (lines 292-296) are hard to understand. The substantial text revision is necessary hopefully with a supplementary figure.

We have intentionally not engaged into an extended discussion of H⁺ coupling as this process has been extensively investigated in prokaryotic transporters^{2,4,11,12}, while the detailed mechanism has still remained elusive (see response to reviewer 2). In our manuscript we thus refer to structural features in DMT1 that are conserved and that have been proposed to play a role in H⁺ transport in previous studies. We intentionally refrained from addressing H⁺ transport in the discussion.

We have reworded the mentioned sentences:

Line 297-303

These acidic residues are in interaction distance to two arginines located on $\alpha 9$ (DMT1: Arg 440, Arg 445), which, due to the poor superposition of this helix compared to the prokaryotic transporters originate from different positions of the helix while their sidechains undergo equivalent interactions to form an extended ionic network (Fig. 5h, Supplementary Figs. 8 and 10d). These residues were previously assigned a role in proton release to the intracellular side with mutations compromising H^+ transport, although the detailed mechanism of proton coupling in the SLC11 family is still debated.

5) Biochemical assays using ion binding-site mutants shown in Fig. 6 need to be conducted with Fe^{2+} as well as Mn^{2+} . This additional data would enable to discuss whether or not the same amino acids are involved in Fe^{2+} binding and transport as Mn^{2+} by DMT1.

We have now included data from the same binding site mutants showing an equivalently strong effect on Fe^{2+} transport as on Mn^{2+} transport (Supplementary Fig. 11c-l). This was expected since previous studies on prokaryotic homologues have already demonstrated that Mn^{2+} and Fe^{2+} occupy the same transport-related binding site¹.

6) Cartoons shown in Fig. 7a are too much simplified. The presented cryo-EM structures provide insight into the TM helix rearrangement during the transition from the occluded to inward/outward-facing states. This information should be included in this figure panel.

We have intentionally provided a simplified representation in Fig. 7a but do not claim that this panel presents a detailed transport mechanism (see response to reviewer 1). The schematic conformational rearrangements are now shown in the novel panel Fig. 7c.

Minor issues:

1) Lines 148-149: The authors state that Sb1 and Sb2 bind to non-overlapping epitopes of DMT1. However, no experimental data are shown to support this conclusion. Some supportive data should be added.

Such data is now shown as Supplementary Fig. 3h.

2) It is very exciting that the authors succeeded in cryo-EM structure analysis of ~52-kDa NRAMP1 without binder proteins although DMT1 with similar size and overall structure to NRAMP1 requires a nanobody for near-atomic resolution cryo-EM analysis. It would be reader-friendly and beneficial for cryo-EM scientists to additionally discuss the reason why the cryo-EM analysis was successful for NRAMP1 in isolation, which is not the case with DMT1.

Our manuscript summarizes over four years of work. We have initially investigated the structure determination of DMT1 at a time when the reconstruction of such small membrane protein without fiducial marker was still considered intractable. As attempts to determine its structure have not yielded high resolution reconstructions, we have decided for the selection of sybodies, which have ultimately allowed us to reach a breakthrough in 3D reconstruction and structure determination at 3.6 Å (Supplementary Fig. 4). Later attempts to determine the structure of the paralog NRAMP1 (which were carried out two years after the structure determination of DMT1), where due to the lower expression and of the membrane protein sybody selection was not an option, have allowed us to obtain the described reconstructions at 3.7 and 3.9 Å respectively. However, despite the nominally similar resolution the map quality in the DMT1 nanobody complex is considerably better. Please also note that the size of the NRAMP1 dataset is much larger (*i.e.* 57'700 micrographs in case of NRAMP1 vs 19'800 in case of DMT1/Sb2). (see also response to reviewer 1)

3) Fig.3c is very unclear. Although the authors point out conformational changes in the extracellular part of alpha6a, the figure panel does not clearly show the location of alpha6a-helix in the whole structure. Figure revision is needed.

The purpose of Fig. 3c is to show that the two structures obtained from different sybody complexes of DMT1 are essentially very similar except for a small conformational difference at the interaction site of Sb1, which is not part of the other complex. Although these differences are small and probably not relevant for protein function, we have revised the figure to better illustrate them.

4) Line 14; “has” should read “have”.

Corrected.

5) Line 76; “our” should read “this”.

Corrected.

6) Line 108; “in DMT1” should read “by DMT1”.

We have reworded this sentence.

7) Line 153; insert “resolutions” after “3.6 Å”.

We have reworded this sentence.

8) Lines 156-157; The phrase “conformational heterogeneity” is very ambiguous. Does the Sb2 itself adopts different conformations or bind to different sites of DMT1?

We have reworded the sentence to: Line 157-159

Whereas both sybodies were visible in certain 2D classes, Sb2, which presumably binds to an intracellular epitope, was not defined after 3D reconstruction due to the intrinsic heterogeneity of the interaction.

9) Line 239; insert “site” after “binding”.

Corrected.

Reviewer #4 (Remarks to the Author):

References

- 1 Ehrnstorfer, I. A., Geertsma, E. R., Pardon, E., Steyaert, J. & Dutzler, R. Crystal structure of a SLC11 (NRAMP) transporter reveals the basis for transition-metal ion transport. *Nat Struct Mol Biol* **21**, 990-996, doi:10.1038/nsmb.2904 (2014).
- 2 Ehrnstorfer, I. A., Manatschal, C., Arnold, F. M., Laederach, J. & Dutzler, R. Structural and mechanistic basis of proton-coupled metal ion transport in the SLC11/NRAMP family. *Nat Commun* **8**, 14033, doi:10.1038/ncomms14033 (2017).
- 3 Manatschal, C. *et al.* Mechanistic basis of the inhibition of SLC11/NRAMP-mediated metal ion transport by bis-isothiourea substituted compounds. *Elife* **8**, doi:10.7554/eLife.51913 (2019).
- 4 Ramanadane, K., Straub, M. S., Dutzler, R. & Manatschal, C. Structural and functional properties of a magnesium transporter of the SLC11/NRAMP family. *Elife* **11**, doi:10.7554/eLife.74589 (2022).
- 5 Ramanadane, K. *et al.* Structural and functional properties of a plant NRAMP-related aluminum transporter. *Elife* **12**, doi:10.7554/eLife.85641 (2023).
- 6 Forbes, J. R. & Gros, P. Iron, manganese, and cobalt transport by Nramp1 (Slc11a1) and Nramp2 (Slc11a2) expressed at the plasma membrane. *Blood* **102**, 1884-1892, doi:10.1182/blood-2003-02-0425 (2003).
- 7 Cunrath, O. & Bumann, D. Host resistance factor SLC11A1 restricts Salmonella growth through magnesium deprivation. *Science* **366**, 995-999, doi:10.1126/science.aax7898 (2019).
- 8 Gunshin, H. *et al.* Cloning and characterization of a mammalian proton-coupled metal-ion transporter. *Nature* **388**, 482-488, doi:10.1038/41343 (1997).
- 9 Bozzi, A. T. *et al.* Conserved methionine dictates substrate preference in Nramp-family divalent metal transporters. *Proc Natl Acad Sci U S A* **113**, 10310-10315, doi:10.1073/pnas.1607734113 (2016).
- 10 Mackenzie, B., Ujwal, M. L., Chang, M. H., Romero, M. F. & Hediger, M. A. Divalent metal-ion transporter DMT1 mediates both H⁺-coupled Fe²⁺ transport and uncoupled fluxes. *Pflugers Arch* **451**, 544-558, doi:10.1007/s00424-005-1494-3 (2006).
- 11 Bozzi, A. T., Bane, L. B., Zimanyi, C. M. & Gaudet, R. Unique structural features in an Nramp metal transporter impart substrate-specific proton cotransport and a kinetic bias to favor import. *J Gen Physiol* **151**, 1413-1429, doi:10.1085/jgp.201912428 (2019).

- 12 Bozzi, A. T. *et al.* Structures in multiple conformations reveal distinct transition metal and proton pathways in an Nramp transporter. *Elife* **8**, doi:10.7554/eLife.41124 (2019).
- 13 Ray, S. *et al.* High-resolution structures with bound Mn(2+) and Cd(2+) map the metal import pathway in an Nramp transporter. *Elife* **12**, doi:10.7554/eLife.84006 (2023).
- 14 Illing, A. C., Shawki, A., Cunningham, C. L. & Mackenzie, B. Substrate profile and metal-ion selectivity of human divalent metal-ion transporter-1. *J Biol Chem* **287**, 30485-30496, doi:10.1074/jbc.M112.364208 (2012).
- 15 Bozzi, A. T., McCabe, A. L., Barnett, B. C. & Gaudet, R. Transmembrane helix 6b links proton and metal release pathways and drives conformational change in an Nramp-family transition metal transporter. *J Biol Chem* **295**, 1212-1224, doi:10.1074/jbc.RA119.011336 (2020).

Response to reviewer requests:

Reviewer #1 (Remarks to the Author):

The authors have improved the description of statistical methods and reporting, and now provide measures of uncertainty for the parameters estimated (e.g. Km, Ki). I appreciate their intent to interpret functional data (for both proteins and mutants) conservatively given the limitations and challenges of their assays. In their rebuttal, the authors state that “The hypothesis of our study is obvious,” and go on to describe the central aim of the study but do not state a hypothesis. To reiterate, the manuscript would benefit from the inclusion of a clearly stated hypothesis in the introduction.

We have introduced a clearly stated hypothesis at the end of the introduction:

Line77-80:

In this study, we were interested in the structural basis of metal ion transport by the human SLC11 paralogs DMT1 and NRAMP1. Particularly, we have addressed the hypothesis that both family members would share general mechanisms with their prokaryotic homologs while displaying unique features, such as the presence of uncoupled H⁺ leaks and the inhibition by Ca²⁺.

Reviewer #3 (Remarks to the Author):

I believe that the authors have addressed all of my concerns appropriately in the revised version. Only minor revisions seem necessary for acceptance of this paper, as described below.

1) Presumed H⁺ transport pathways in DMT1 are illustrated in newly prepared Fig. 5H. Although this added figure is highly beneficial for readers, the left panel displaying the overall structure is still hard to grasp due to too small size, too small and imprecisely positioned labels, and the shaded square. It would better that the shaded square is replaced by a blank square, and that residues involved in the H⁺ transport are highlighted in the right inset with an outer frame line.

We have revised Fig. 5H accordingly.

2) In supplementary Fig. 11c-l, the labels “Mn²⁺” on the right should be amended to “Fe²⁺”.

We have introduced the corrections in Supplementary Fig. 11c-l.

In response to requests by the editor we have introduced the following changes:

Line 115-117:

Although only providing indirect evidence, this result suggests that the two metal ions could compete for the same binding site and consequently slow down Mn^{2+} transport (Fig. 1e).

Line 249-254:

A group of negatively charged residues in DMT1 (Glu 120, Glu 283 and Asp 469), would provide favorable electrostatics for attracting divalent metal ions and could presumably be part of an inhibitory Ca^{2+} binding site, which would stabilize the observed conformation and thus account for the described non-competitive inhibition of transport at high extracellular Ca^{2+} concentrations^{1,2} that was also observed in our experiments, although this assignment is at this stage hypothetical (Figs. 1f, 4f, Supplementary Fig. 1j).

Line 377-378:

However, this proposal is at this stage hypothetical and still requires experimental validation.